# Intracellular localisation of *Mycobacterium tuberculosis* affects efficacy of the antibiotic pyrazinamide

Pierre Santucci[1], Daniel J. Greenwood [1,4], Antony Fearns[1], Kai Chen[2], Haibo Jiang [2,3✉] & Maximiliano G. Gutierrez [1✉]

To be effective, chemotherapy against tuberculosis (TB) must kill the intracellular population of the pathogen, *Mycobacterium tuberculosis*. However, how host cell microenvironments affect antibiotic accumulation and efficacy remains unclear. Here, we use correlative light, electron, and ion microscopy to investigate how various microenvironments within human macrophages affect the activity of pyrazinamide (PZA), a key antibiotic against TB. We show that PZA accumulates heterogeneously among individual bacteria in multiple host cell environments. Crucially, PZA accumulation and efficacy is maximal within acidified phagosomes. Bedaquiline, another antibiotic commonly used in combined TB therapy, enhances PZA accumulation via a host cell-mediated mechanism. Thus, intracellular localisation and specific microenvironments affect PZA accumulation and efficacy. Our results may explain the potent in vivo efficacy of PZA, compared to its modest in vitro activity, and its critical contribution to TB combination chemotherapy.

[1] Host–Pathogen Interactions in Tuberculosis Laboratory, The Francis Crick Institute, London, UK. [2] School of Molecular Sciences, University of Western Australia, Perth, AU, Australia. [3] Department of Chemistry, The University of Hong Kong, Hong Kong, China. [4] Present address: Institute of Molecular Systems Biology, ETH, Zurich, Switzerland. ✉email: hbjiang@hku.hk; max.g@crick.ac.uk

**M**ycobacterium tuberculosis (Mtb), the etiologic agent of tuberculosis (TB), is the most prevalent cause of mortality due to an infectious agent worldwide[1]. Drug-sensitive TB requires treatment with a minimum of four anti-biotics over a course of at least 6 months[1]. The standard first-line multidrug therapy includes the drugs isoniazid (INH), rifampicin (RIF), ethambutol and pyrazinamide (PZA). The duration and toxicity of the current anti-TB regimens affect compliance, leading to treatment failure, relapse and emergence of resistance. The increasing number of multidrug-resistant strains constitutes a global health issue, and new therapeutic strategies are needed to reduce the treatment duration of drug-sensitive TB. However, this is challenging in the context of TB chemotherapy, since the environments within tuberculous granulomas are heterogeneous and dynamic[2,3].

Mtb is a pathogen able to infect, adapt, survive and replicate in several cell types within its human host. During pulmonary infection, Mtb faces multiple environments, where alveolar macrophages constitute a key replication niche in early infection and thus play a central role in the tubercle bacilli lifecycle[4]. The intracellular lifestyle of Mtb represents a crucial stage in TB pathogenesis and it is now accepted that drug discovery pro-grammes should include in cellulo studies using infected host cells[5–8]; which may be more accurate than classical in vitro-based screens using only isolated bacteria. Within macrophages, Mtb encounters diverse subcellular environments including both membrane-bound compartments, such as phagosomes, phagoly-sosomes and autophagic compartments, and the cytosol[9–11]. These environments exhibit distinct biophysical and biochemical properties such as nutrient availability, pH, and hydrolytic activities that can affect Mtb replication[12,13]. In this context, macrophage activation enhances anti-mycobacterial activities and changes Mtb sensitivity towards first-line drugs[14]. Despite its physiological relevance, if bacterial compartmentalisation alters drug accumulation and efficacy remains poorly characterised. Moreover, how specific subcellular microenvironments impact antibiotic mode of action is elusive.

The contribution of intracellular environments to antibiotic efficacy is particularly important for antitubercular compounds such as the front-line drug PZA; that is mainly effective in vivo but has very little potency in vitro[15–18]. The activity of PZA against Mtb was discovered in the 1950s using TB mouse models before being further investigated in drug-susceptible TB patients[19]. The remarkable efficacy of PZA played a key role in shortening the anti-TB chemotherapy from 9 to 6 months[20]. PZA synergises with new drugs such as pretomanid or bedaquiline (BDQ) and therefore it is included in new generations of shorter and less toxic regimens[21–24]. Some of these combinations sup-ported by the Global Alliance for TB Drug Development are currently under phase III clinical development, including the trials *STAND* and *SimpliciTB*[25,26].

Although PZA is widely used, the contrasting differences between its in vitro and in vivo efficacy is still not fully under-stood. Pioneering studies demonstrated that PZA is inactive in vitro at neutral pH but displays activity against Mtb at pH 5.5 or below[16,27,28]. PZA is a pro-drug that requires conversion by the mycobacterial nicotinamidase/pyrazinamidase PncA into pyrazinoic acid (POA), which is subsequently exported to the extracellular milieu. If this environment is acidic, POA becomes protonated (HPOA) and crosses the bacterial envelope to finally accumulate within Mtb[29,30]. This pH-dependent model remains a commonly accepted mechanism of PZA accumulation where HPOA is subsequently able to disrupt membrane potential and decrease intrabacterial pH[28,31]. However, PZA pH-independent activity have also been reported and additional PZA/POA modes of action proposed[32–34]. For example, PZA also inhibits the biosynthesis of Coenzyme-A through inhibition of the aspartate decarboxylase PanD and enhancement of its degradation by the Clp protease system[35–37]. Thus, the molecular mechanisms responsible for PZA efficacy are complex and diverse. Notably, most of the mechanistic studies are performed in vitro and do not consider the intracellular lifestyle of Mtb.

Here, we used a recently developed correlative light, electron and ion microscopy (CLEIM) approach to visualise antibiotics at a subcellular resolution[38] and investigate PZA distribution in human primary macrophages infected with Mtb. By combining high-content microscopy with genetic and pharmacological per-turbations, we analysed how changes in Mtb intracellular locali-sation and host-cell environments affect PZA localisation, accumulation and efficacy. We provide evidence that acidic pH is an important factor underlying PZA enrichment and efficacy in cellulo. Finally, by imaging the subcellular distribution of two different anti-TB antibiotics for the first time, we show that BDQ enhances PZA accumulation in intracellular mycobacteria through a host-dependent mechanism. Our results show that efficacy may be affected by the localisation of bacteria at the time of treatment and provide new directions for the design of future antibiotics and combined therapies.

## Results

**PZA/POA localises primarily within intracellular Mtb.** To define the subcellular distribution of PZA, human blood monocyte-derived macrophages (MDM) were infected with Mtb and analysed by correlative electron and ion microscopy (Sup-plementary Fig. 1)[38]. Previous studies showed that the average PZA maximum serum concentration ($C_{max}$) in PZA-treated TB patients is circa 30.8 mg/L[39]. In order to mimic this physiological concentration and better define PZA distribution in cellulo, MDM were infected with Mtb for 24 h and further treated with 30 mg/L of [$^{15}N_2$, $^{13}C_2$]-labelled PZA for an additional 24 h. Because both isotopic labels are present on the pyrazine ring, they are preserved when PZA is converted to POA by the bacterial enzyme PncA (Supplementary Fig. 1). Consequently, ion micro-scopy cannot differentiate between PZA and POA or other metabolites, which will collectively be referred here as PZA/POA. PZA/POA was detectable by the enrichment of $^{15}N$ relative to $^{14}N$ in treated samples (Supplementary Fig. 1, Fig. 1) but not by the $^{13}C$ label when compared to control samples, indicating that a higher concentration of $^{13}C$ might be required to be detectable by ion microscopy, due to higher natural abundance of $^{13}C$ (1.1%) compared to the natural abundance of $^{15}N$ (0.37%), and $^{13}C$ background signals from the resin. The $^{31}P$ signals, derived mainly from nucleic acids and polyphosphates structures were used to discriminate intracellular bacteria (Supplementary Fig. 1, Fig. 1). We observed that $^{15}N/^{14}N$ ratio was only above natural abundance within intracellular Mtb but not in the host-cell cytosol or any other macrophage organelles (Fig. 1a). To dis-criminate whether this accumulated form was the pro-drug PZA or its active form POA, we used in this experimental system two strains from the *M. tuberculosis* complex that are unable to convert PZA into POA. Both *Mycobacterium bovis* and *Myco-bacterium bovis BCG* strains possess a non-functional *pncA* gene which prevents both strains to generate POA[40,41]. Quantitative analysis showed that no positive $^{12}C^{15}N$ enrichment derived from the isotopically labelled-PZA was detectable within both strains, suggesting that conversion of PZA into POA by PncA is essential for $^{12}C^{15}N$ accumulation (Supplementary Fig. 2). Consequently, our results suggest that the main accumulated form of the drug within intracellular bacteria is likely to be its active form POA.

Interestingly, we notice that some infected macrophages were totally devoid of PZA/POA enrichment (Fig. 1b). Also, in some

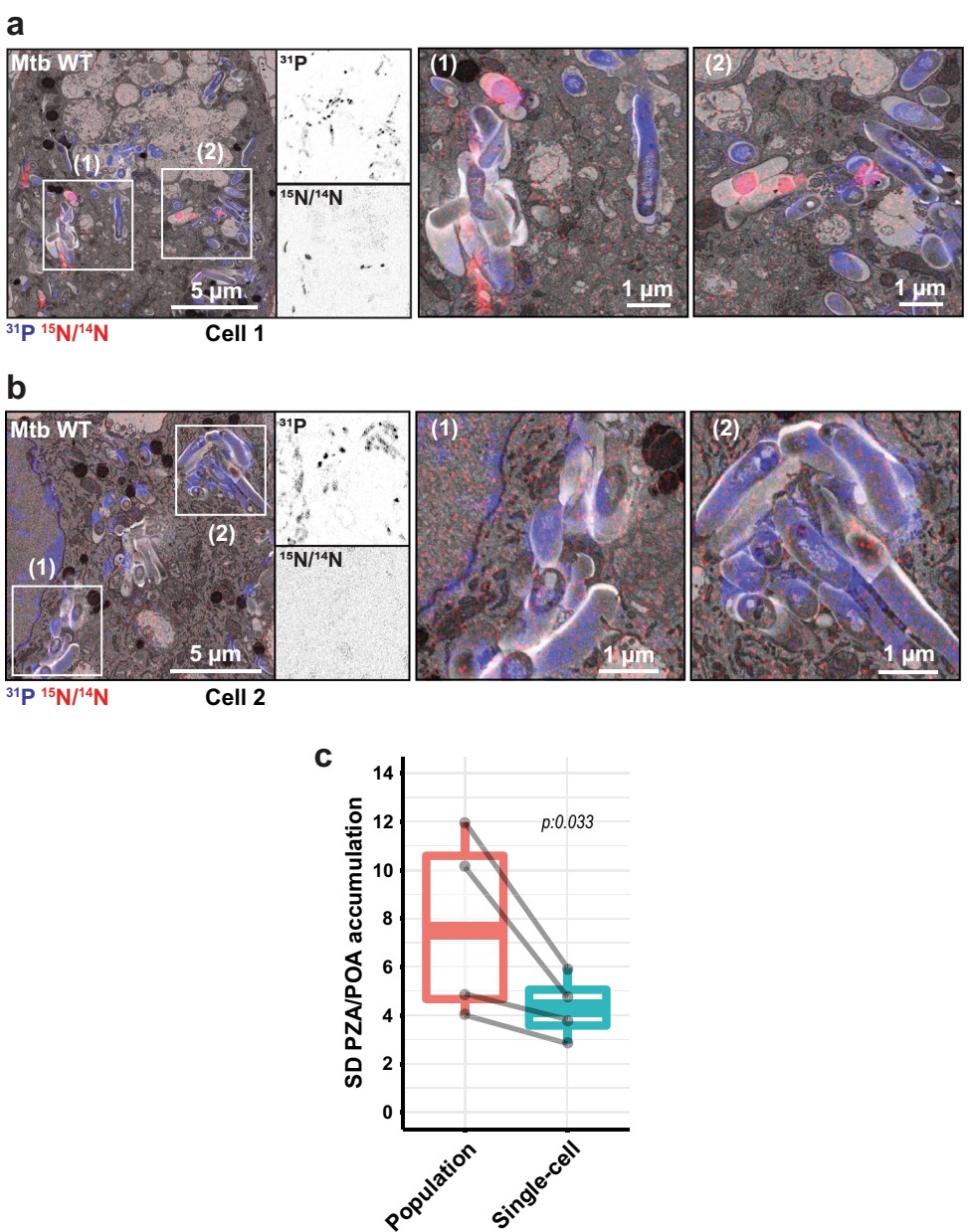

**Fig. 1 Heterogeneous accumulation of PZA/POA within Mtb-infected MDM. a, b** Mtb-infected MDM treated with 30 mg/L [$^{15}$N$_2$, $^{13}$C$_2$]-PZA for 24 h. EM micrograph is overlaid with $^{31}$P (blue) and $^{15}$N/$^{14}$N (red) NanoSIMS images. Magnifications show $^{31}$P (top panel) and $^{15}$N/$^{14}$N (bottom panel) individual signals at the single bacterial-cell level. Scale bar corresponds to 5 μm. Regions of interest (1) and (2) highlighted by white rectangles, are shown in detail in the right panels respectively. Scale bar corresponds to 1 μm. Micrographs are representative of four independent experiments. **c** Comparison of standard deviation (SD) in $^{15}$N/$^{14}$N ratiometric signals between bacteria contained within the same macrophage (single-cell level) and across Mtb in all macrophages (population level). A total of 1649 intracellular bacteria were analysed from $n = 4$ biologically independent experiments. Results are shown as box-and-whisker plot where minima, maxima, median and percentile are displayed with the linked-results from individual experiments. Statistical significance was assessed by using a two-tailed t-statistic test from the linear model. Source data are provided as Source Data file.

PZA/POA positive cells the accumulation was highly heterogeneous, even between neighbouring bacteria (Fig. 1a). This suggested that each individual host macrophage may have effects on the bacterial PZA/POA accumulation. To test this hypothesis, the distribution of PZA/POA enrichment between bacteria within each single macrophage was compared to the distribution of enrichment across the infected macrophage population (Fig. 1c). Consistent with the hypothesis, the intra-macrophage standard deviation of accumulation was found to be statistically lower than at the whole-population level (Fig. 1c). Moreover, analysis of PZA/POA enrichment within Mtb contained in intact cells (Fig. 1) or necrotic cells revealed that plasma membrane integrity

positively impacts drug accumulation (Supplementary Fig. 3). Given that loss of membrane integrity in necrotic cells dissipates pH gradients, this finding is consistent with a pH-dependent accumulation of PZA. Thus, PZA/POA accumulation in the context of infection is highly heterogenous between host cells and also between intracellular Mtb.

**The PZA/POA accumulation in Mtb is affected by the intra-cellular localisation.** Because low pH has been shown to enhance PZA/POA enrichment and activity under certain culture conditions, we next analysed using ion microscopy whether Mtb

subcellular localisation could affect antibiotic distribution and/or accumulation within host cells.

We first used a genetic approach by comparing Mtb WT, which can localise in the cytosol, with the Mtb ΔRD1 mutant that lacks the ESX-1 Type 7 secretion system and resides primarily in membrane-bound compartments[10,11,42]. We postulated that the restriction of ESX-1 deficient-Mtb to membrane-bound compartments would increase the proportion of bacteria in acidic microenvironments, as previously described[43,44], and thus might impact PZA/POA accumulation. Co-localisation analysis of Mtb WT and Mtb ΔRD1-infected cells stained with the acidotropic dye LysoTracker confirmed that Mtb ΔRD1 was more often associated with LysoTracker positive environments (Supplementary Fig. 4). In addition, to exclude any intrinsic variation in PZA susceptibility between Mtb WT and Mtb ΔRD1, a drug susceptibility assay was performed in vitro. Our results were in line with previously published data regarding the weak efficacy of PZA in vitro in neutral broth[27] and further confirmed that there were no PZA dose-response differences in the relative growth between the two strains (Supplementary Fig. 5a). Moreover, no differences in $^{15}$N/$^{14}$N signal were observed between Mtb WT and Mtb ΔRD1 when incubated for 24 h in the presence of [$^{15}$N$_2$, $^{13}$C$_2$]-labelled PZA in vitro (Supplementary Fig. 5b). A quantitative analysis of PZA/POA accumulation in Mtb-infected macrophages revealed a higher level of antibiotic associated with Mtb ΔRD1 than the Mtb WT strain (average mean intensity per positive bacteria of 93.02 ± 4.98 vs. 65.77 ± 3.27, $p$:2.6 × 10$^{-8}$) (Fig. 2a–c), consistent with the proposal that membrane-bound compartments promote PZA/POA accumulation. Analysis of the distribution of PZA/POA positive bacteria between Mtb WT and Mtb ΔRD1 by density plot confirms that WT bacteria harbouring PZA/POA positive signature displayed a lower enrichment than ΔRD1 bacteria (Fig. 2e). In addition to a significant increase in $^{15}$N/$^{14}$N ratio per bacteria, the percentage of PZA/POA positive bacteria was also higher during infection with the Mtb ΔRD1 strain (Fig. 2d).

Next, we postulated that if the pH-dependent model of PZA accumulation is correct, inhibition of endosomal acidification would result in reduced enrichment of the antibiotic within intracellular bacteria. To test this hypothesis, macrophages were infected with either Mtb WT or Mtb ΔRD1 and further treated with [$^{15}$N$_2$, $^{13}$C$_2$]-PZA in the absence or presence of the pharmacological modulator Bafilomycin-A1 (BafA1), a specific inhibitor of the mammalian vacuolar-type H$^+$-ATPase[45]. We first assessed BafA1 activity and confirmed that treatment for 2, 24 or 72 h significantly reduced pH-dependent proteolytic activity of macrophages without affecting cell viability (Supplementary Figs. 6, 7). As expected, BafA1 treatment of Mtb-infected cells for 24 h resulted in a significant diminution of bacteria in acidic compartments (Supplementary Fig. 4). The level of PZA/POA in treated macrophages was still heterogenous with few bacteria that strongly accumulated PZA/POA whereas many others were negative for $^{15}$N/$^{14}$N enrichment (Fig. 2f, g). Quantitative analysis of intrabacterial $^{15}$N/$^{14}$N ratios in the presence or absence of BafA1 revealed that inhibition of endolysosomal acidification significantly reduced PZA/POA level in Mtb ΔRD1 (average mean intensity per positive bacteria of 93.02 ± 4.98 vs. 78.5 ± 5.93, $p < 0.05$) but were similar in Mtb WT (average mean intensity per positive bacteria of 65.77 ± 3.27 vs. 69.19 ± 7.48, $p$:0.36) when compared to their respective untreated control samples (Fig. 2h, c). Moreover, pairwise comparison between Mtb WT and Mtb ΔRD1 upon BafA1 treatment showed no significant differences in mean $^{15}$N/$^{14}$N signal and percentage of PZA/POA positive bacteria ($p$:0.44 and $p$:0.67, respectively). That this decrease occurs primarily with Mtb ΔRD1 upon endolysosomal

acidification inhibition argues that Mtb restricted to membrane-bound compartments are more susceptible to PZA/POA accumulation. Altogether, these results support a pH-dependent model of PZA accumulation within intracellular Mtb, highlighting that the subcellular localisation of the tubercle bacilli and the nature of the microenvironments it encounters within human macrophages impact PZA accumulation.

**The intracellular localisation of Mtb affects PZA efficacy.** Next, we investigated if these changes in bacterial intracellular localisation and PZA/POA enrichment had an effect on efficacy. The effect of PZA on intracellular Mtb replication was assessed by using a high-content single-cell analysis imaging approach[38] (Supplementary Fig. 8). After 96 h of infection, Mtb WT replicated faster than the Mtb ΔRD1 mutant in resting human macrophages (Fig. 3a, b) as previously reported[10,42]. Replication inhibition profiles observed with Mtb WT at 30- and 100 mg/L agreed with previous results obtained in PZA-treated human macrophages after 96 h of infection showing that PZA is efficient in cellulo[46]. In contrast to our in vitro results, Mtb ΔRD1 was more susceptible to PZA than Mtb WT in cellulo (Fig. 3a, b). Indeed, analysis of total bacterial levels at 72 h post-treatment in comparison to the 24 h pre-treatment values demonstrated that PZA was more effective against Mtb ΔRD1 when compared with Mtb WT (Fig. 3b). A relative growth inhibition analysis showed that at 30 mg/L, PZA is more effective against Mtb ΔRD1 than Mtb WT (Fig. 3e). Interestingly, at a higher concentration (100 mg/L), the inhibition was more efficient but the difference between the Mtb WT and Mtb ΔRD1 was reduced, suggesting that at higher concentrations PZA might distribute more evenly into more intracellular microenvironments.

Then, we tested if modulation of endolysosomal acidification affects PZA efficacy. For this, macrophages were infected with Mtb WT or Mtb ΔRD1 for 24 h and then treated with PZA in presence or absence of the vacuolar-type H$^+$-ATPase inhibitor BafA1 (therefore an inhibitor of endolysosomal acidification). Treatment with BafA1 clearly increased both Mtb WT and Mtb ΔRD1 replication in macrophages (Fig. 3c, d and Supplementary Fig. 9), confirming that acidification/proteolytic activity is mostly restrictive for Mtb at some stages of the intracellular lifestyle[47]. Similar results were obtained with Concanamycin A (ConA), another vacuolar-type H$^+$-ATPase inhibitor[48] (Supplementary Fig. 9). Neither BafA1 nor ConA had a direct effect on Mtb WT or Mtb ΔRD1 growth in vitro (Supplementary Fig. 9). BafA1 treatment reduced PZA efficacy when used at 30 mg/L resulting in a significantly higher bacterial burden, with changes in inhibition values from 40% to 20% for Mtb WT and from 70% to 25% for the Mtb ΔRD1 mutant (Fig. 3f). These findings were also assessed independently with an alternative technique and many similarities were observed by colony-forming units (CFU) bulk analysis (Supplementary Fig. 10). Indeed, inhibition of the replication of Mtb ΔRD1 mutant was more important than its WT counterpart when treated with 30 mg/L of PZA (Supplementary Fig. 10). In addition, the BafA1 antagonistic effect towards PZA was also more pronounced in Mtb ΔRD1 than Mtb WT strain supporting the findings obtained by high-content fluorescence microscopy analysis (Supplementary Fig. 10). Interestingly, the greater effect of BafA1 on Mtb ΔRD1, which are mostly in a membrane-bound/acidic environment, clearly supports a role for low pH in PZA/POA efficacy. However, the exposure to BafA1 for 24 h or longer often led to a recovery phenotype that resulted in partial inhibition of lysosomal activity (Supplementary Figs. 4, 5). Co-localisation between Mtb and LysoTracker in the presence of ConA revealed that this v-ATPase

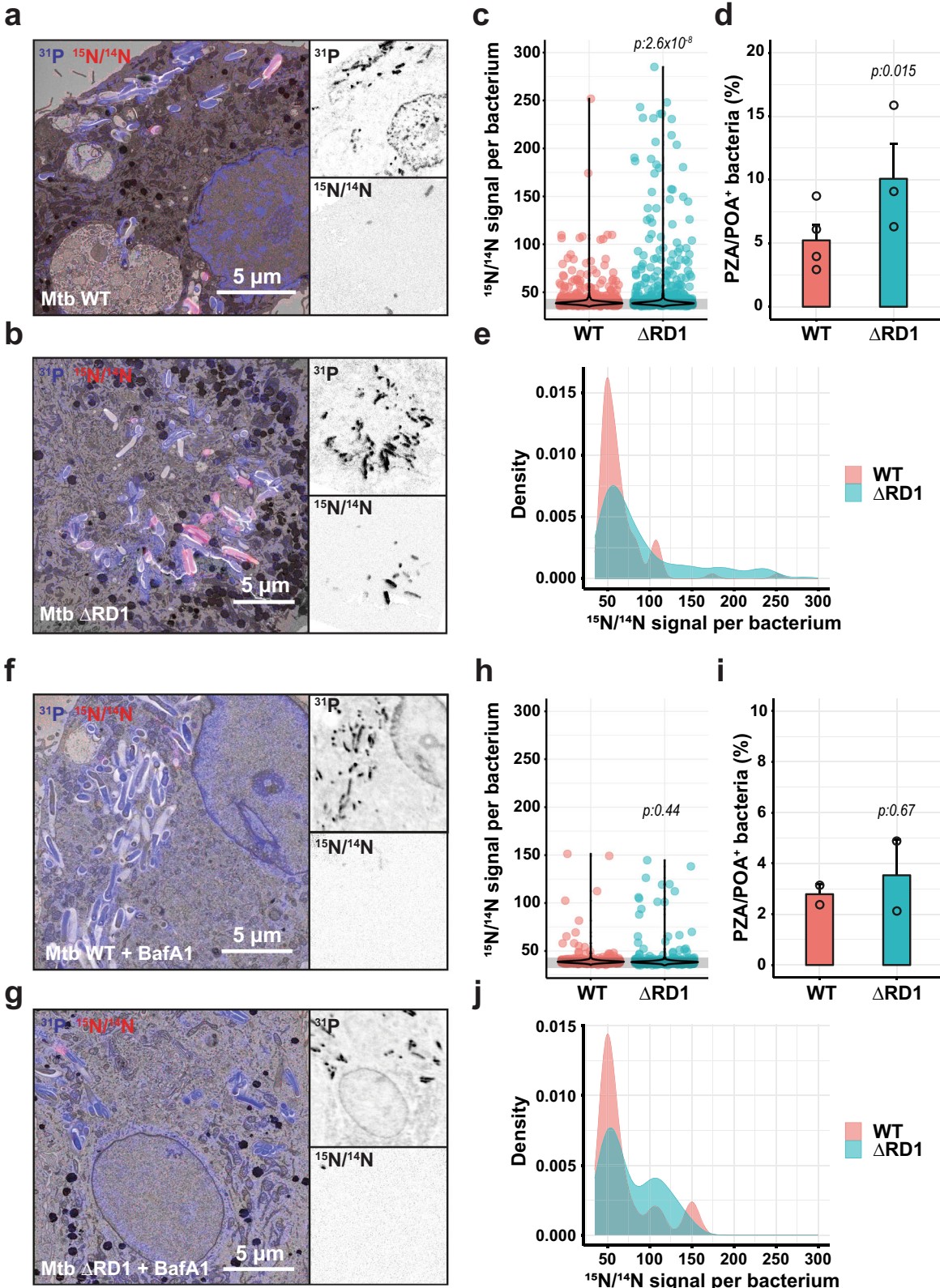

inhibitor was more efficient than BafA1 specially during longer exposure times (Supplementary Fig. 4). Thus, we conducted a complementary set of experiments with this inhibitor. ConA inhibited endolysosomal acidification and proteolytic activity as measured by LysoTracker and DQ-BSA labelling (Supplementary Fig. 11) as well as PZA/POA accumulation which was impaired in both intracellular Mtb WT and Mtb ΔRD1 (Supplementary

Fig. 11). Intracellular antibiotic activity assays performed in the presence of ConA, resulted in an almost complete suppression of PZA inhibitory effect (Supplementary Fig. 11). Importantly, this antagonistic effect observed in the presence of modulators was specific for PZA, since endolysosomal acidification inhibition by pharmacological modulation did not significantly affect the efficacy of RIF or INH (Supplementary Fig. 12). Altogether,

**Fig. 2 Subcellular localisation and endolysosomal acidification contribute to PZA/POA accumulation in intracellular Mtb. a, b** Representative images of PZA/POA distribution in intracellular Mtb WT and Mtb ΔRD1 strains. MDM were infected with Mtb WT or Mtb ΔRD1 and treated with 30 mg/L [$^{15}N_2$, $^{13}C_2$]-PZA for 24 h. EM micrographs are overlaid with $^{31}P$ (blue) and $^{15}N/^{14}N$ (red) NanoSIMS images. Magnifications show $^{31}P$ (top panel) and $^{15}N/^{14}N$ (bottom panel) individual images at the single bacterial-cell level. Scale bars correspond to 5 μm. Micrographs are representative of 3–4 independent experiments. **c** Quantitative analysis of $^{15}N/^{14}N$ signal per bacterium shown as violin plot with single dots. Grey line indicates the natural background level of the $^{15}N/^{14}N$ enrichment. Results were obtained from 1352 to 1649 individually segmented bacteria from $n = 3$ or 4 biologically independent experiments and $p$ values were calculated by using a two-tailed t-statistic test from the linear model. **d** Quantification of the percentage of PZA/POA positive (PZA/POA$^+$) bacteria. Results are expressed as mean ± SEM from $n = 3$ or 4 biologically independent experiments and $p$ values were calculated by using a two-tailed t-statistic test from the linear model. **e** Analysis of the $^{15}N/^{14}N$ signal profile of Mtb WT and Mtb ΔRD1 PZA/POA positive bacteria. **f, g** Representative images of PZA/POA distribution in intracellular Mtb WT and Mtb ΔRD1 strains treated with 30 mg/L [$^{15}N_2$, $^{13}C_2$]-PZA for 24 h in the presence of 100 nM BafA1. Micrographs and magnifications are displayed as described for (**a, b**). Scale bars correspond to 5 μm. Micrographs are representative of 2–3 independent experiments. **h** Quantitative analysis of $^{15}N/^{14}N$ signal per bacterium shown as violin plot with single dots. Grey line indicates the natural background level of the $^{15}N/^{14}N$ enrichment. Results were obtained from 741 to 828 individually segmented bacteria from $n = 2$ or 3 biologically independent experiments and $p$ values were calculated by using a two-tailed t-statistic test from the linear model. **i** Quantification of the percentage of PZA/POA positive (PZA/POA$^+$) bacteria after 24 h of 100 nM BafA1 treatment. Results are expressed as mean ± SEM from $n = 2$ or 3 biologically independent experiments and $p$ values were calculated by using a two-tailed t-statistic test from the linear model. **j** Analysis of the $^{15}N/^{14}N$ signal profile of Mtb WT and ΔRD1 PZA/POA positive bacteria in the presence of 100 nM BafA1. All $p$ values were considered significant when $p \leq 0.05$. Source data are provided as Source Data file.

these data provide evidence that endolysosomal function and acidification is critical for PZA/POA enrichment and inhibitory activity, highlighting that a pH-dependent mechanism contributes to PZA/POA accumulation and efficacy in cellulo.

**PZA/POA accumulates in Mtb residing in non-acidic compartments.** Our findings suggested that intracellular pH is an important factor contributing to PZA/POA accumulation and efficacy in cellulo. To further explore whether distinct intracellular populations of Mtb accumulate PZA/POA in a pH-dependent manner, we used a CLEIM approach. Macrophages were infected with fluorescent Mtb, treated with 30 mg/L of [$^{15}N_2$, $^{13}C_2$]-PZA for 24 h and acidic compartments were labelled using LysoTracker. Within single macrophages, we analysed the PZA/POA enrichment in both LysoTracker positive and LysoTracker negative Mtb (Fig. 4). We observed that most of the intracellular bacteria analysed within a single macrophage were harbouring extremely low levels of PZA/POA almost similar to the natural background enrichment ($^{15}N/^{14}N$ signal values ranging from 37 to 44) regardless of the pH microenvironment, as measured by LysoTracker staining. Strikingly, two bacteria contained in a LysoTracker negative compartment accumulated higher levels of PZA/POA (Fig. 4), suggesting that PZA/POA enrichment can potentially occur in non-acidic environments in our experimental model. These observations suggest that PZA/POA pH-independent mechanism of accumulation might take place in cellulo. However, it is worth mentioning that residence in LysoTracker positive environments is dynamic and fluctuates rapidly in the context of Mtb infection in macrophages[42,49]. Moreover, chemical fixation could have altered LysoTracker signal. These findings highlight that despite being a powerful technology, CLEIM might require to be associated with high-content live-imaging approach to fully catch the spatiotemporal dynamics of Mtb infection and finely dissect the intracellular pharmacokinetics of antibiotics. Altogether, these data revealed that intrabacterial PZA/POA accumulation within host cells is a dynamic process which might explain the highly heterogeneous enrichment of both Mtb WT and Mtb ΔRD1.

**BDQ enhances the accumulation of PZA in intracellular Mtb.** In macrophages, BDQ enhances the targeting of Mtb to acidic compartments and synergises with PZA[50]. We hypothesised that BDQ-dependent targeting to acidic compartments will increase PZA enrichment within intracellular Mtb. BDQ did not increase

PZA/POA enrichment per bacterium when co-treated at both neutral and acidic pH in vitro (Supplementary Fig. 13). In contrast, BDQ significantly enhanced PZA/POA accumulation in intracellular Mtb (Fig. 5a, b). To determine if there is a direct correlation between the amount of BDQ per bacterium and the increase in PZA/POA, we performed a quantitative analysis of both BDQ and the PZA/POA levels per single-bacterium ($R = 0.14$, $p:0.035$). With a coefficient correlation within 0–0.3 range and a significant p value[51], these results indicate that there was no linear association between BDQ and the PZA/POA signal. This agrees with previous reports suggesting that BDQ triggers host-cell-mediated changes that impacts PZA accumulation and efficacy[23,50]. To analyse synergistic effects due to bacterial fitness or viability impairment, we repeated these experiments in the presence of two non-labelled antibiotics. Mtb WT-infected MDM were co-treated with labelled-PZA and the front-line drugs RIF or INH, which inhibit Mtb growth in cellulo without directly impacting host-activation status, and PZA/POA accumulation was further determined by NanoSIMS (Supplementary Fig. 14). RIF has been previously reported to synergise with PZA in vivo[52,53]. Consistent with these observations, we found that RIF co-treatment increased intrabacterial $^{15}N/^{14}N$ levels and the proportion of PZA/POA positive bacteria (~37% of positive bacteria with an average mean intensity of 87.95 ± 2.91). In contrast, quantitative analysis of $^{15}N/^{14}N$ enrichment upon INH co-treatment did not show any effect when compared to PZA treatment alone (~7% of positive bacteria with an average mean intensity of 56.03 ± 2.36) (Supplementary Fig. 14). These results argue that synergy can occur via distinct processes, including host-mediated and host-independent mechanisms, and also suggest that antibiotic-mediated growth restriction does not necessarily lead to increased levels of PZA/POA.

During PZA-BDQ co-treatment, we also observed that high intrabacterial PZA/POA levels had a particular spatial distribution within intracellular Mtb. High-resolution ion micrographs of $^{31}P$, $^{79}Br/^{12}C^{14}N$ and $^{15}N/^{14}N$ signal analysis and 3D surface plot reconstruction showed that PZA/POA signal was mainly detected at the periphery of the bacterial cell suggesting a strong association with the mycobacterial plasma membrane, the cell wall or potentially the surrounding extrabacterial environment (Fig. 5c, d). In contrast, the $^{79}Br/^{12}C^{14}N$ signal corresponding to the BDQ signal was diffuse through the bacterial cytosol (Fig. 5c).

We then tested if the effect of BDQ on PZA accumulation is reversed by inhibition of endolysosomal acidification. Human macrophages were infected with Mtb WT and incubated with labelled PZA and BDQ in the presence or absence of 100 nM

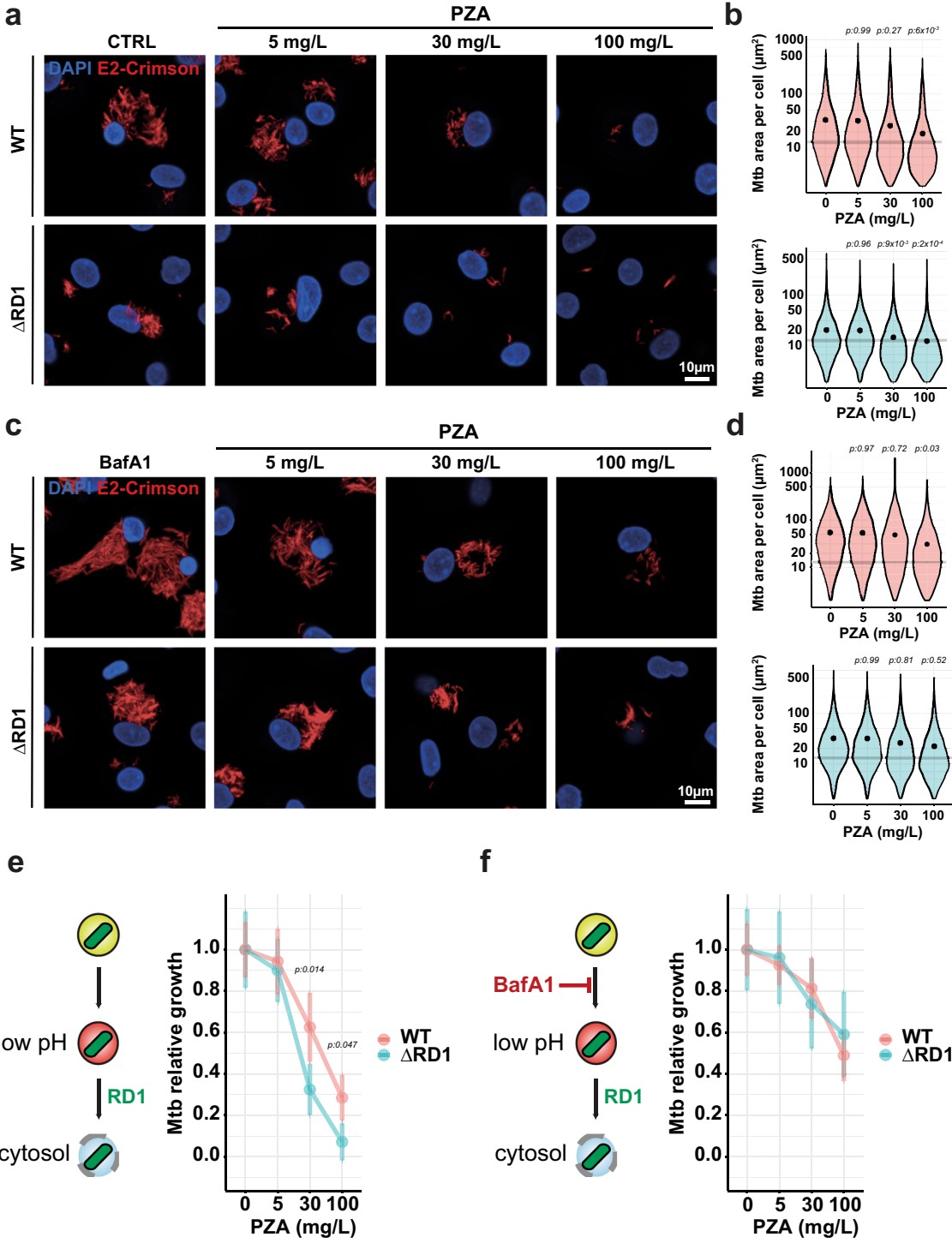

BafA1. The inhibition of endolysosomal acidification decreased PZA/POA levels associated with Mtb (Fig. 5e–g) (average mean intensity per positive bacteria of $138.84 \pm 2.27$ vs. $82.19 \pm 1.75$, $p:2.2 \times 10^{-14}$). Moreover, the percentage of positive bacteria upon BafA1 treatment decreased ~1.6-fold (Fig. 5h). Confirming these observations, distribution of the $^{15}N/^{14}N$ enrichment of the population significantly changed after BafA1 treatment (Fig. 5i). Altogether, by simultaneously imaging two antibiotics, we show that BDQ enhances PZA accumulation within Mtb through endolysosomal targeting, and that the synergistic effect observed could be partially counteracted by inhibiting the activity of vacuolar-type H$^+$-ATPase.

## Discussion

By using correlative imaging of antibiotics at the subcellular level, here we provide evidence supporting the idea that distinct antibiotics will differentially accumulate in intracellular compartments with direct consequences for efficacy. The front-line anti-TB drug PZA distributed very differently from BDQ, which was shown to accumulate in host-cell lipid droplets[38]. PZA/POA primarily localises in intracellular Mtb and not in any other organelles or the cytosol, though we cannot rule out that some PZA/POA was washed out during sample processing. PZA/POA was only detectable by the enrichment of $^{15}N$ relative to $^{14}N$ in treated samples but not by the $^{13}C$ label, likely due to matrix effects.

**Fig. 3 PZA inhibits Mtb replication in cellulo and is more effective against Mtb ΔRD1 mutant strain. a** Representative confocal fluorescence images of Mtb WT- and Mtb ΔRD1-infected MDM for 24 h and further treated for 72 h with increasing concentration of PZA. Magnifications display nuclear staining (blue) and Mtb-producing E2-Crimson (red). Scale bar corresponds to 10 μm. Micrographs are representative of four independent experiments. **b** Quantitative analysis of E2-Crimson Mtb WT (top panel) and Mtb ΔRD1 (bottom panel) area per single-cell expressed in μm$^2$. Results are displayed in violin plots where grey lines represent the mean Mtb area per cell pre-treatment (t$_{24h}$) and black dots represent the mean Mtb area per cell post-treatment (t$_{96h}$). From 7124 to 8785 and 5419 to 6560 infected MDM were analysed for Mtb WT and Mtb ΔRD1 respectively. Results were obtained from $n = 4$ biologically independent experiments and statistical significance was assessed by comparing the means of each conditions using one-way ANOVA followed with Tukey's multiple comparisons test. **c** Representative confocal fluorescence images of Mtb WT- and Mtb ΔRD1-infected MDM for 24 h and further treated for 72 h with increasing concentration of PZA in the presence of 100 nM BafA1. Scale bar corresponds to 10 μm. Micrographs are representative of 4 independent experiments. **d** Quantitative analysis of E2-Crimson Mtb WT (top panel) and Mtb ΔRD1 (bottom panel) area per single-cell expressed in μm$^2$, is represented as in (**b**). From 7821 to 9130 and 6619 to 7723 infected MDM were analysed for Mtb WT and Mtb ΔRD1 respectively. Results were obtained from $n = 4$ biologically independent experiments and statistical significance was assessed by comparing the means of each conditions using one-way ANOVA followed with Tukey's multiple comparisons test. **e, f** Mean bacterial area per macrophage in the presence or absence of 100 nM BafA1 and increasing concentration of PZA was normalised and plotted as relative growth. A schematic representation of the BafA1 effect onto endolysosomal pH accompanied the graphs. Normalisation was done to mean Mtb area per cell pre-treatment (t$_{24h}$) and the control condition without PZA was used as reference corresponding to 100% growth. Results obtained from $n = 4$ biologically independent experiments are displayed as mean ± SEM and statistical significance was assessed between Mtb WT and Mtb ΔRD1 by comparing the means of Mtb relative growth for each condition using one-way ANOVA followed by a pairwise $t$ test. All $p$ values were considered significant when $p \leq 0.05$. Source data are provided as Source Data file.

The $^{15}$N isotopic label used during PZA synthesis was positioned in the pyrazine ring and did not allow discriminating between PZA and POA by ion microscopy in this study. Our experiments with *M. bovis*, the etiologic agent of bovine TB and the vaccine strain *M. bovis* BCG strongly suggest that the signal we detect by NanoSIMS is mostly from POA molecules. These two strains, which belong to the *M. tuberculosis* complex, encode a *pncA* gene identical to Mtb *pncA* with the exception of a well-characterised C169G single-nucleotide polymorphism. This mutation is responsible for the His57Asp substitution which is detrimental for PncA enzymatic activity and prevents both strains of converting PZA into its active form POA. Pyrazinamidase deficiency results in a complete abolition of $^{15}$N/$^{14}$N enrichment, suggesting that the signal detected within intracellular Mtb WT and Mtb ΔRD1 throughout our study is likely to be the active form POA. Our findings in cellulo are in agreement with the pioneer work performed with radiolabelled-PZA by Zhang et al. who demonstrated that POA, but not PZA, mainly accumulates within Mtb in vitro[28].

Given that PZA is an effective antitubercular drug in vivo, we expected that it would homogenously accumulate in Mtb-infected macrophages at drug concentrations close to the $C_{max}$. However, we observed that not all macrophages had detectable levels of PZA/POA. The possibility that sample processing can differentially affect macrophages in the same population is very low. In this context, very little is known about the mechanisms that allow internalisation of antibiotics in infected cells[54,55] and we postulate that the phenotypic state of individual cells determines drug uptake and enrichment. There was also a highly heterogenous distribution of Mtb-associated PZA/POA signal, strongly suggesting that at physiological levels, PZA/POA does not accumulate in all intracellular Mtb.

Our results highlight the contribution of single-host-cell and single-bacterial cell heterogeneity in infection outcome but more importantly in antibiotic efficacy. Uncovering how host phenotypic and metabolic features might positively or negatively impact drug levels and subsequently antimicrobial efficacy against intracellular pathogens has become of great interest. In this context, experimental systems that combine high-content single-cell quantitative fluorescence microscopy with genetically labelled host-microenvironments and bacterial reporters will greatly improve our understanding of antibiotic modes of action against intracellular bacteria. These experimental approaches might also be valuable during the early steps of antimicrobial discovery and development pipelines.

Interestingly, we reported that in necrotic cells, where permeability of host membranes is higher, very few bacteria were enriched with PZA/POA which agrees with in vivo studies[23,56,57]. Although PZA/POA distributes across necrotic tissue, the biophysical parameters of these specific lesions, such as neutral pH, has been proposed to be a key factor linked to PZA/POA inefficacy against Mtb in mouse and guinea pig models[56,57].

In our experimental system, a significant proportion of intracellular Mtb is cytosolic at the time of treatment, which suggests that the neutral pH of cytosol could be detrimental for PZA efficacy. On the other hand, membrane-bound restriction renders Mtb ΔRD1 more susceptible to PZA. It is likely that ESX-1-mediated phagosomal membrane damage affects the pH surrounding Mtb[43]. Indeed, Mtb harbouring mutations in the ESX-1-encoding genes fail to prevent phagosome acidification. Furthermore, Mtb mutants in the *phoPR* two-component system, which blocks the secretion of EsxA[58,59], were localised in acidic phagosomes. Pharmacological inhibition of the ESX-1 secretion system results in more Mtb localising in acidic compartments[44]. Thus, ESX-1-mediated secretion reduces exposure of Mtb to acidic compartments and might explain the higher accumulation of PZA/POA within intracellular Mtb ΔRD1 compared to cytosolic Mtb WT. We cannot exclude that other factors potentially different between Mtb WT and Mtb ΔRD1 might be responsible for these differences. Given that side-by-side comparison experiments in vitro did not show major changes in susceptibility between Mtb WT and Mtb ΔRD1, it is likely that the host-cell microenvironment is the main driver of the ESX-1 dependent alterations in PZA accumulation.

Phagosome acidification is also required for PZA/POA accumulation in intracellular Mtb, suggesting that this antibiotic acts by a host cell-driven pH-dependent mechanism. Changes in pH or proteolytic activity can modulate drug efficacy in mouse macrophages[14] and our results indicate that acidification and proteolytic activity contributes to Mtb restriction in human macrophages as well. This agrees with data showing that inhibition of endolysosomal acidification increases mycobacterial replication[47,60]. Importantly, both Mtb ESX-1-dependent localisation and phagosome acidification are required for efficient PZA efficacy, as shown by intracellular antibiotic susceptibility assays. Inhibitors of the vacuolar-type H$^+$-ATPase drastically impacted PZA accumulation and efficacy in our biological system. This supports the idea that the main form accumulated is likely POA, since preventing accumulation result in loss of antibiotic efficacy. We also noticed that BafA1 long-term exposure was associated with slow recovery of lysosomal function/acidification which may lead to intermediate phenotypes. In some conditions where

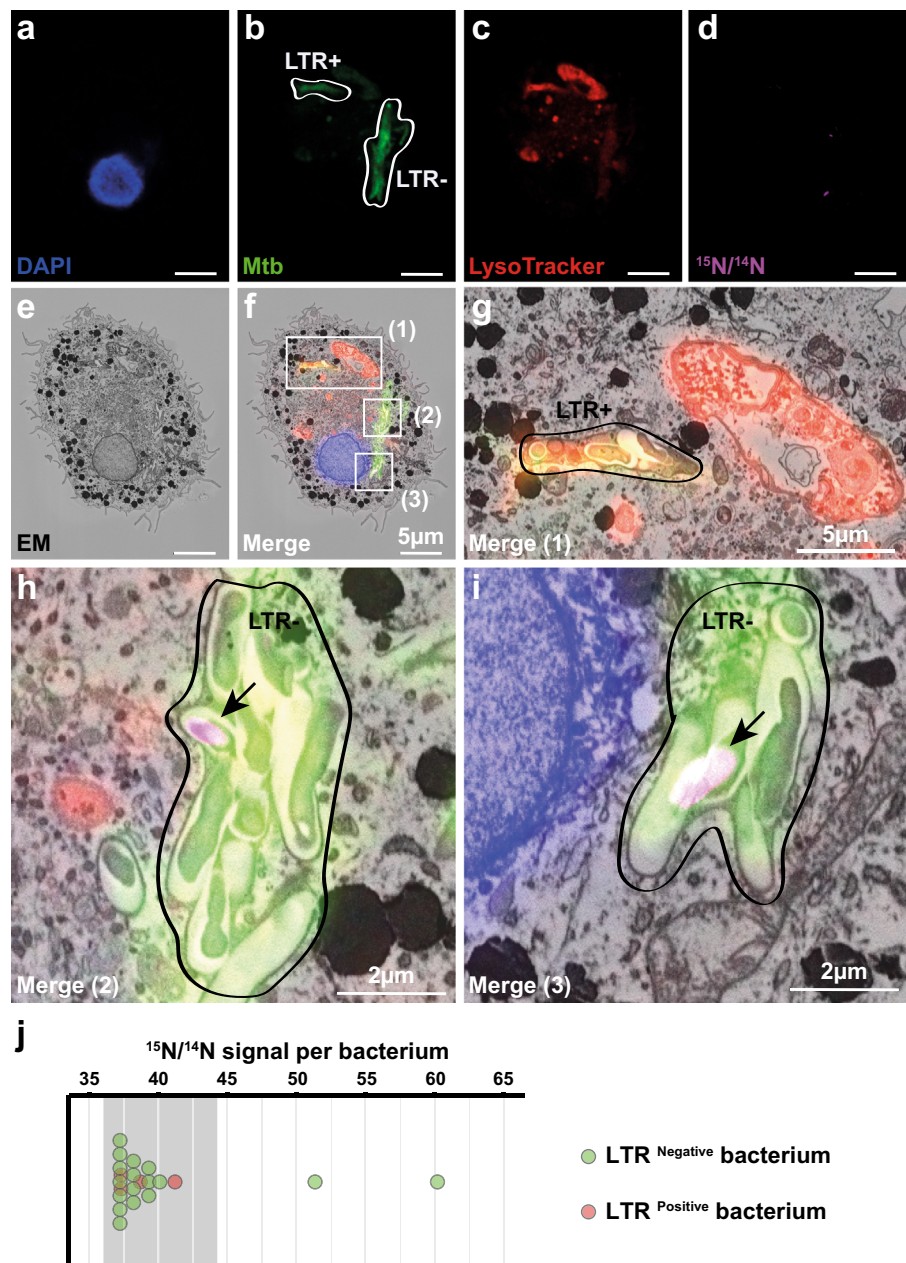

**Fig. 4 CLEIM reveals that PZA/POA is able to accumulate within Mtb residing in non-acidic compartments. a–i** Mtb-infected MDM were treated with 30 mg/L [$^{15}N_2$, $^{13}C_2$]-PZA for 24 h. The sample was then stained with 200 nM LysoTracker before chemical fixation and counter-stained with DAPI. After confocal fluorescence imaging, sample was embedded and the cell of interest was processed for correlative electron and ion microscopy. Representative fluorescence images corresponding to (**a**) nuclear staining (blue), (**b**) Mtb WT (green), (**c**) LysoTracker (red) were overlaid with (**d**) the $^{15}N/^{14}N$ signal (purple) and (**e**) the EM micrograph (black and white). Merging of the five channels is shown in (**f**) where three regions of interest numbered (1), (2) and (3), respectively, are highlighted by white rectangles. Scale bars correspond to 5 µm. Micrographs are representative of one single experiment. **g** Magnification of region (1) shows bacteria contain within LTR positive compartment devoid of any detectable PZA/POA. Scale bar corresponds to 5 µm. **h, i** Magnifications of regions (2) and (3) highlight PZA/POA accumulation in LTR negative microenvironment. Scale bar corresponds to 2 µm. **j** Quantification of $^{15}N/^{14}N$ signals associated with each bacterium contained in LysoTracker positive and negative microenvironments. LysoTracker positivity was determined by measuring the mean fluorescence values from the two compartments of interest. Diffuse signal from the cytosol was estimated between 10 and 20 AU and considered as background, whereas mean values were 22.6 and 60.4 AU for negative and positive compartments respectively. Grey line indicates $^{15}N/^{14}N$ natural background level. Bacterium with strong positive $^{15}N/^{14}N$ signals correspond to the bacterium highlighted in (2) and (3) respectively. Source data are provided as Source Data file.

acidification was partially impaired, PZA was still able to restrict Mtb WT and Mtb ΔRD1 replication in a dose dependent manner. ConA was more potent in the long-term and allowed to almost completely rescue Mtb WT and ΔRD1 growth upon PZA treatment. This prevention of PZA efficacy was significantly reduced with both inhibitors of acidification, and this rescuing effect was more pronounced in the membrane-bound Mtb ΔRD1 strain. We hypothesise that both cytosolic access and restriction to membrane-bound non-acidic environment may provide protective niches against PZA/POA. Recent studies performed in vitro have shown that PZA is efficacious in non-acidic conditions[61,62]. Nevertheless, in conditions where acidification is impaired, the

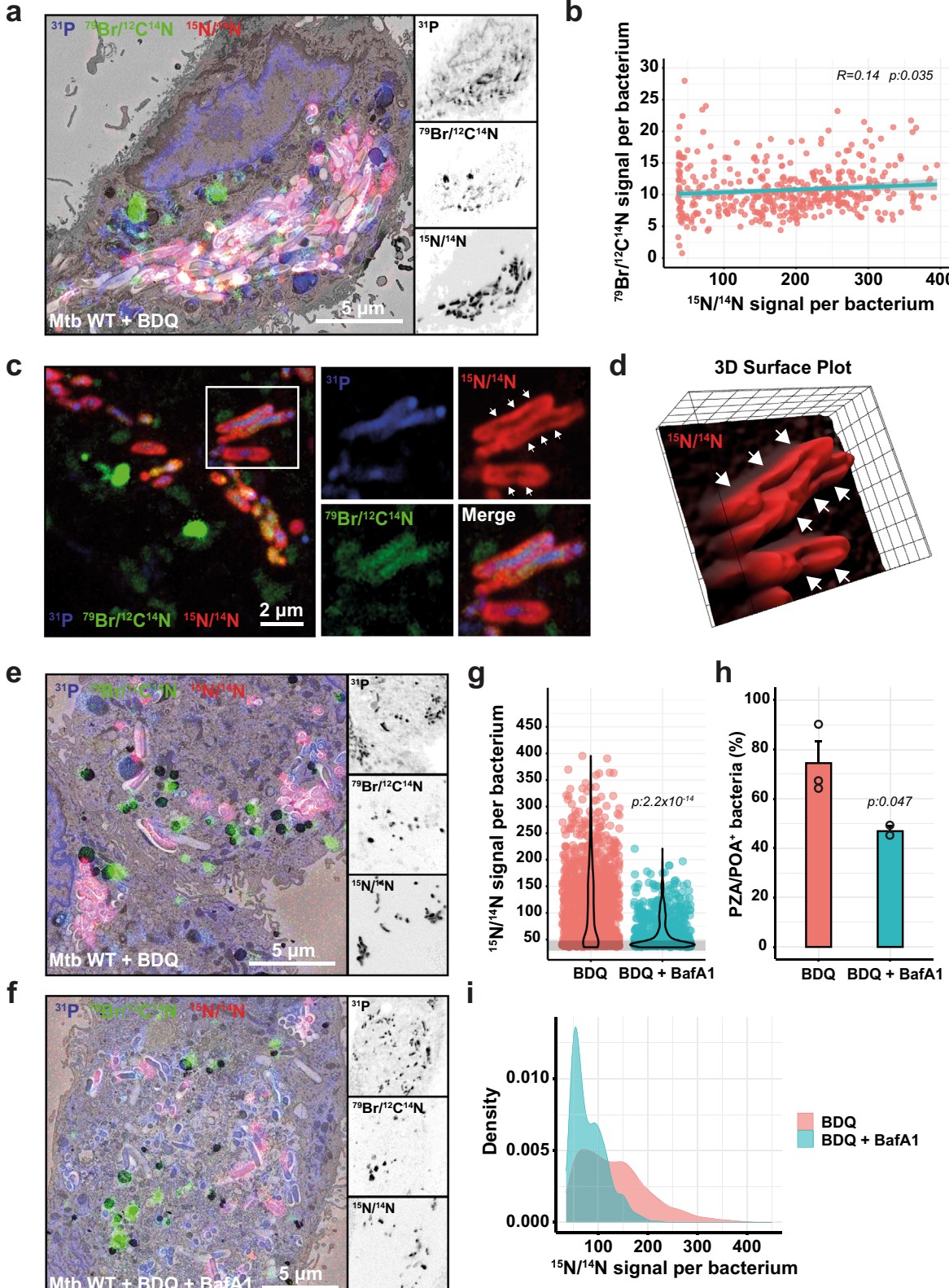

PZA ability to restrict Mtb WT and Mtb ΔRD1 replication is limited suggesting that intracellular pH is likely a major contributor of PZA/POA activity in cellulo.

By using a correlative approach that enabled imaging of both PZA and host acidic microenvironments, we discovered that PZA can accumulate in bacteria contained in non-acidic compartments. However, the dynamics of Mtb infection in human macrophages is complex[42,49] and it will be important to define at which stage the accumulation of the antibiotic happens with higher temporal resolution. Further studies will define whether time of residence in specific microenvironments define drug enrichment and efficacy. Other bacterial factors that are only expressed within host cells can potentially affect PZA accumulation in acidic and non-acidic environments need to be

**Fig. 5 Dual subcellular antibiotic visualisation shows that BDQ enhances PZA accumulation within Mtb-infected MDM. a** Mtb-infected MDM treated with 30 mg/L [$^{15}N_2$, $^{13}C_2$]-PZA and 2.5 mg/L of BDQ for 24 h. EM micrograph is overlaid with $^{31}P$ (blue), $^{79}Br/^{12}C^{14}N$ (green) and $^{15}N/^{14}N$ (red) NanoSIMS images. Magnifications show $^{31}P$ (top panel), $^{79}Br/^{12}C^{14}N$ (middle panel) and $^{15}N/^{14}N$ (bottom panel) individual signals at the single bacterial-cell level. Scale bar corresponds to 5 μm. Micrographs are representative of two independent experiments. **b** Pearson's correlation between $^{15}N/^{14}N$ (x-axis) and $^{79}Br/^{12}C^{14}N$ (y-axis) signals in individual bacterium. Results are from 414 intracellular Mtb WT obtained from two biological replicates. Correlation coefficient (R) shown as the cyan line and the corresponding p value was calculated using a correlation coefficient table based on the degrees of freedom and assed by two-tailed t test. Micrographs are representative of two independent experiments. **c** NanoSIMS images of Mtb-infected MDM treated with 30 mg/L [$^{15}N_2$, $^{13}C_2$]-PZA and 2.5 mg/L of BDQ for 24 h. Images were filtered with a median filter using the OpenMIMS plugin in FIJI and pseudo-colours are displayed as in (**a**). Magnifications show the region of interest highlighted by white rectangle. White arrows indicate the peripheral signal of PZA/POA (**d**) 3D surface plot of the $^{15}N/^{14}N$ signal from the bacteria depicted in (**c**) was generated using the 3D surface plot from FIJI. White arrows show the PZA/POA accumulation in 3D. **e**, **f** Mtb-infected MDM treated with 30 mg/L [$^{15}N_2$, $^{13}C_2$]-PZA and 2.5 mg/L BDQ in the presence or absence of 100 nM BafA1 for 24 h. EM micrograph and magnifications are displayed as in (**a**). Micrographs are representative of 2–3 independent experiments. **g** Quantitative analysis of $^{15}N/^{14}N$ signal per bacterium treated with BDQ alone or in combination with 100 nM BafA1 shown as violin plot with single dots. Grey line indicates the natural background level of the $^{15}N/^{14}N$ enrichment. Results were obtained from 810 to 1275 individually segmented bacteria from $n = 2$ or 3 biologically independent experiments and p values were calculated by using a two-tailed t-statistic test from the linear model. **h** Quantification of the percentage of PZA/POA positive (PZA/POA$^+$) bacteria. Results are expressed as mean ± SEM from $n = 2$ or 3 biologically independent experiments and p values were calculated by using a two-tailed t-statistic test from the linear model. **i** Analysis of the $^{15}N/^{14}N$ signal profile of Mtb WT PZA/POA positive bacteria treated with BDQ alone or in combination with 100 nM BafA1. Source data are provided as Source Data file.

considered. For example, intracellular bacterial fitness, metabolism and drug influx/efflux activity might influence antibiotic accumulation and efficacy. Altogether, our data show that the heterogeneity of PZA enrichment in Mtb is likely the result of a combination of both bacterial and host-cell factors.

BDQ, an antibiotic that targets the mycobacterial ATP synthase and also displays antibacterial activity by increasing lysosomal activity, significantly increased PZA/POA accumulation in Mtb. We did not observe any correlation between the accumulation of BDQ and PZA/POA per single-bacterium suggesting that the two drugs may have a cooperative mechanism of action as previously suggested[23]. Moreover, there was no synergistic effect in vitro either at neutral or acidic pH, arguing that the effect of BDQ is host-mediated as previously shown[50]. Both our in vitro and in cellulo experiments suggest that BDQ-induced reduction of ATP levels, does not impair active influx/efflux of PZA/POA, however further experiments are required to confirm these findings.

RIF but not INH co-treatment increases the levels of intrabacterial PZA/POA, supporting previous observations reporting in vivo synergy[52,53]. These results highlight that our analytical pipeline is useful to understand the molecular and cellular bases of drug-drug interactions. Bacterial viability or fitness impairment upon INH-mediated mycolic-acid biogenesis inhibition does not correlate with higher PZA/POA enrichment, suggesting that antibiotic synergy requires in some cases host-cell activation and lysosomal function whereas others do not. Accordingly, inhibition of BDQ-dependent lysosomal acidification partially reduces the proportion of PZA/POA positive intracellular mycobacteria. This argues that host-mediated mechanisms triggered by BDQ are likely pH-driven and can be counteracted by pharmacological modulation. Host-cell activation by cytokine or pharmacological induction has been previously reported to impact antibiotic efficacy in mouse macrophages[14,63]. Based on these studies, it is tempting to hypothesise that interferon stimulation or activation of the transcriptional factor EB will result in microenvironment changes with consequences for antibiotic efficacy[14,63].

Finally, the simultaneous subcellular imaging of two structurally unrelated antibiotics allowed us to define intrabacterial distribution of the drugs. Unexpectedly, we found that PZA/POA spatial distribution within individual bacteria was primarily associated with the periphery of the bacillus. Although the biological significance is unclear, it is tempting to postulate that such accumulation is either due to the localisation of the bacteria within a tight phagosome or linked to the biological target(s) of the PZA/POA. Protonated POA is able to accumulate and further disrupt membrane potential, membrane transport and intrabacterial pH[31]. Recent structural and biochemical studies, clearly demonstrated that POA molecules are also able to covalently bind the Mtb protein PanD in vitro[35,64]. Although the bacterial localisation of PanD is unknown, it is possible that CoA biosynthesis enzyme might localise at the inner leaflet of the bacterial cytoplasmic membrane to meet the requirement of specific lipid biosynthesis pathways as it was previously shown between the FAS-II and the mycolic-acid synthesis machineries[65].

Thus, host-cell environments and Mtb intracellular localisation affect antibiotic efficacy, arguing that different antibiotics are needed to target heterogeneous intracellular bacterial populations. These results provide an explanation for the observed sterilising activity of PZA in the clinic but also contributes to a conceptual framework for a host-cell dependent combined drug therapy in the context of TB. It is likely that many other antibiotics used for the treatment of intracellular pathogens display localisation-dependent potency and efficacy.

## Methods

**Bacterial strains and growth conditions.** *M. tuberculosis* H37Rv WT and ΔRD1[66] strains were obtained from William R. Jacobs Jr. (Albert Einstein College of Medicine, New-York, USA), Suzie Hingley-Wilson (University of Surrey, Guilford, UK) and Douglas Young (The Francis Crick Institute, London, UK). Fluorescent *M. tuberculosis* H37Rv pTEC19 (Mtb WT) and *M. tuberculosis* H37Rv ΔRD1 pTEC19 (Mtb ΔRD1) strains were used in this study[42]. Recombinant strains harbouring pTEC19 plasmid (Addgene, #30178) and producing the fluorescent protein E2-Crimson were grown in Middlebrook 7H9 broth supplemented with 0.2% glycerol (v/v) (Fisher Chemical, G/0650/17), 0.05% Tween-80 (v/v) (Sigma-Aldrich, P1754) and 10% ADC (v/v) (BD Biosciences, 212352). Both strains were previously tested for PDIM positivity by thin layer chromatography of lipid extracts from Mtb cultures[42]. Hygromycin B (Invitrogen, 10687010) was used at a concentration of 50 mg/L as a selection marker for the fluorescent strains. Bacterial cultures (10 mL) were incubated at 37 °C with rotation in 50 mL conical tubes.

**Antibiotic susceptibility testing in broth.** Susceptibility testing of Mtb WT and Mtb ΔRD1 was performed by end-point determination of optical density at 600 nm (OD$_{600nm}$) using the broth microdilution method in 96-well clear flat-bottom microplates (Corning, 353072). Both Mtb WT and Mtb ΔRD1 strains were grown in complete Middlebrook 7H9 broth until mid-exponential phase (OD$_{600nm} = 0.6 \pm 0.2$) and further adjusted to a bacterial density of $\sim 5 \times 10^6$ bacteria/mL (OD$_{600nm} = 0.05$) assuming that an OD$_{600nm}$ of 1 approximates to $10^8$ bacteria/mL. A volume of 100 μL of inoculum was then added to each well containing 100 μL of twofold serial dilutions of PZA (Sigma, P7136) ranging from 4 to 512 mg/L in complete Middlebrook 7H9 broth, thus giving a final bacterial load of $\sim 2.5 \times 10^6$ bacteria/mL (OD$_{600nm} = 0.025$). In addition, wells containing 200 μL of 7H9 medium only were used as sterility/background controls, whereas wells containing 5 mg/L or 2.5 mg/L of RIF (LKT laboratories, R3220) and BDQ (MedChemExpress, HY-14881) respectively, were used as positive controls for growth inhibition. Plates were sealed in zipped-lock bags and incubated at 37 °C without

agitation. After 14 days, plates were scanned and $OD_{600nm}$ was determined using a microplate reader (Berthold, Mithras[2] LB 943). Results were expressed as relative growth where Mtb WT and Mtb ΔRD1 growth in complete Middlebrook 7H9 broth was considered as 100%. Experiments were performed in biological duplicate with at least six technical replicates.

**Preparation of human-monocyte derived macrophages.** Human MDM were prepared from Leucocyte cones (NC24) supplied by the NHS Blood and Transplant service[10,38]. White blood cells were isolated by centrifugation on Ficoll-Paque Premium (GE Healthcare, 17-5442-03) for 60 min at $300 \times g$. Mononuclear cells were collected and washed twice with MACS rinsing solution (Miltenyi, 130-091-222) to remove platelets and red blood cells. The remaining were lysed by incubation with 10 mL RBC lysing buffer (Sigma, R7757) per pellet for 10 min at room temperature. Cells were washed with rinsing buffer and then were re-suspended in 80 μL MACS rinsing solution supplemented with 1% BSA (Miltenyi, 130-091-376) (MACS/BSA) and 20 μL anti-CD14 magnetic beads (Miltenyi, 130-050-201) per $10^8$ cells. After 20 min on ice, cells were washed in MACS/BSA solution and re-suspended at a concentration of $10^8$ cells/500 μL in MACS/BSA solution and further passed through an LS column (Miltenyi, 130-042-401) in the field of a QuadroMACS separator magnet (Miltenyi, 130-090-976). The LS column was washed three times with MACS/BSA solution, then CD14 positive cells were eluted, centrifuged and re-suspended in complete RPMI 1640 with GlutaMAX and HEPES (Gibco, 72400-02), 10% foetal bovine serum (Sigma, F7524) and 10 ng/ml of hGM-CSF (Miltenyi, 130-093-867). Cells were plated at a concentration of $10^6$ cells/mL in untreated petri dishes. Dishes were incubated in a humidified 37 °C incubator with 5% $CO_2$. After 3 days, an equal volume of fresh complete media including hGM-CSF was added. Six days after the initial isolation, differentiated macrophages were detached in 0.5 mM EDTA in ice-cold PBS using cell scrapers (Sarsted, 83.1830), pelleted by centrifugation and re-suspended in RPMI medium containing 10% foetal bovine serum where cell count and viability was estimated (BioRad, TC20™ Automated Cell Counter) before plating for experiments.

**Macrophage infection with Mtb.** For macrophage infection, Mtb WT and Mtb ΔRD1 inoculum were prepared by centrifuging approximately 10 mL of mid-exponential phase bacterial cultures ($OD_{600nm} = 0.6 \pm 0.2$) which were further washed twice in sterile PBS buffer (pH 7.4)[10,38]. Then, an equivalent volume of sterile 2.5–3.5 mm autoclaved glass beads was added to the pellet and bacterial clumps were disrupted by vigorously shaking. Bacteria were re-suspended in cell culture media and the remaining clumps were removed by slow-speed centrifugation at $300 \times g$ for 5 min. The supernatant was transferred to a fresh tube and $OD_{600nm}$ measured to determine bacterial concentration, assuming once again that an $OD_{600nm}$ of 1 approximates to $10^8$ bacteria/mL. For intracellular antibiotic assays, macrophages were infected with Mtb WT and Mtb ΔRD1 at a multiplicity of infection (MOI) of 1 for 2 h at 37 °C. For ion microscopy and full-correlative experiments a MOI of 5-10 was used. After 2 h of uptake, cells were washed twice with PBS to remove extracellular bacteria and fresh media was added.

**Intracellular antibiotic activity assays in Mtb-infected macrophages.** Intracellular antibiotic activity assays were performed by high-content fluorescence quantitative imaging[38]. Briefly, $3.5 \times 10^4$ cells per well were seeded into an olefin-bottomed 96-well plate (Perkin Elmer, 6055302) 16–20 h prior to infection. Cells were infected as described above for 24 h and the culture media was replaced by fresh media containing increasing concentrations of PZA, RIF, INH or left untreated. When indicated, fresh medium containing 100 nM Bafilomycin-A1 (BafA1) (Enzo Life Sciences, BML-CM110-0100) or 100 nM Concanamycin A (ConA) (Sigma-Aldrich, C9705) was added together with the antibiotics. At the required time point, infected cells were washed twice with PBS buffer (pH 7.4) and fixed with a 4% methanol-free paraformaldehyde (Electron Microscopy Sciences, 15710) in PBS buffer (pH 7.4) for 16–20 h at 4 °C. Fixative was removed and cells were washed two times in PBS buffer (pH 7.4) before performing DAPI (Invitrogen, D1306) staining for nuclear visualisation. Image acquisition was performed with the OPERA Phenix high-content microscope with a ×40 water-immersion 1.1 NA objective. The confocal mode with default autofocus and a binning of 1 was used to image multiple fields of view (323 μm × 323 μm) from each individual well with 10% overlapping, where acquisition was performed at 4 distinct focal planes spaced with 1 or 2 μm. DAPI stained nuclei were detected using $\lambda_{ex} = 405$ nm/$\lambda_{em} = 450$ nm, where the laser power was set at 20% with an exposure time of 200 ms. E2-Crimson bacteria were detected using $\lambda_{ex} = 595$ nm/$\lambda_{em} = 633$ nm, the laser power was set at 30% with an exposure time of 200 ms. Analysis was performed using the Harmony software (Perkin Elmer, version 4.9) where maximum projection of the four z-planes was used to perform single-cell segmentation by using the 'Find nuclei' and 'Find cells' building blocks. Cells on the edges were excluded from the analysis. The E2-Crimson signal was detected using the 'Find Image Region' building block where a manual threshold was applied to accurately perform bacterial segmentation. The Mtb area per cell was determined by quantifying the total area (expressed in μm²) of E2-Crimson⁺ signal per single macrophage. The relative growth inhibition was determined by using the following formula ~ (Mean Mtb area per cell $t_{96 h}$ − Mean Mtb area per cell $t_{24 h}$)/(Mean Mtb area per cell $t_{24 h}$) and the relative values were obtained by using the untreated

control as a reference of 100% growth (0% inhibition). All the results were exported as CSV files, imported in the R studio software (The R Project for Statistical Computing, version 1.3.1073) and graphs were plotted with the ggplot2 package (version 3.3.2).

**Resin embedding and block trimming.** Mtb-infected cells were washed once in HEPES 0.2 M buffer, fixed with 2.5% glutaraldehyde (Sigma, G5882), 4% methanol-free paraformaldehyde (Electron Microscopy Sciences, 15710) in 0.2 M HEPES (pH 7.4) for 16–20 h at 4 °C and further processed for Scanning Electron Microscopy (SEM) and nanoscale secondary ion mass spectrometry (NanoSIMS) in a Biowave Pro (Pelco, USA) with use of microwave energy and vacuum. Cells were washed twice in HEPES 0.2 M buffer (Sigma-Aldrich, H0887) at 250 W for 40 s, post-fixed using a mixture of 2% osmium tetroxide (Taab, O011) and 1.5% potassium ferricyanide (Taab, P018) (v/v) at equal ratio for 14 min with power set up at 100 W (with/without vacuum 20″ Hg at 2 min intervals). Samples were washed with distilled water twice on the bench and twice again in the Biowave 250 W for 40 s and further stained in 1% thiocarbohydrazide (Sigma-Aldrich, 223220) in distilled water (w/v) for 14 min as described above. After repeated washes, samples were also stained with 2% osmium tetroxide (Taab, O011) in distilled water (w/v) for 14 min with similar settings. Samples were washed 4 times as before and stained with 1% aqueous uranyl acetate (Agar scientific, AGR1260A) in distilled water (w/v) for 14 min and then washed again. Samples were dehydrated using a step-wise ethanol series of 50, 75, 90 and 100%, then washed four times in absolute acetone at 250 W for 40 s per step. Samples were infiltrated with a dilution series of 25, 50, 75 and 100% Durcupan ACM® (Sigma-Aldrich, 44610) (v/v) resin to acetone. Each step was for 3 min at 250 W power (with/without vacuum 20″ Hg at 30 s intervals). Samples were then cured for a minimum of 48 h at 60 °C. The sample block was trimmed, coarsely by a razor blade then finely trimmed using a 35° ultrasonic, oscillating diamond knife (DiATOME, Switzerland) set at a cutting speed of 0.6 mm/s, a frequency set by automatic mode and a voltage of 6.0 V, on an ultramicrotome EM UC7 (Leica Microsystems, Germany) to remove all excess resin surrounding an area for analysis.

For bacterial pellets, fixed bacteria were pelleted at $3000 \times g$ for 5 min and washed twice in HEPES 0.2 M buffer (Sigma-Aldrich, H0887) at 250 W for 40 s, post-fixed using a mixture of 2% osmium tetroxide (Taab, O011) and 1.5% potassium ferricyanide (Taab, P018) (v/v) at equal ratio for 14 min with power set up at 100 W (with/without vacuum 20″ Hg at 2 min intervals). Samples were washed twice with distilled water in the Biowave at 250 W for 40 s and further stained with 1% aqueous uranyl acetate (Agar scientific, AGR1260A) in distilled water (w/v) for 14 min and then washed again as before. Between steps, bacteria were centrifuged at 3000 g for 5 min. Samples were dehydrated using a step-wise ethanol series of 50, 75, 90 and 100%, then washed four times in absolute acetone at 250 W for 40 s per step. Samples were infiltrated with a dilution series of 25, 50, 75 and 100% Durcupan ACM® (Sigma-Aldrich, 44610) (v/v) resin to acetone. Each step was for 3 min at 250 W.

**Nanoscale secondary ion mass spectrometry (NanoSIMS).** For ion microscopy experiments, cells were infected for 24 h and the culture media was replaced by fresh media containing either 30 μg/mL [$^{15}N_2$, $^{13}C_2$]-PZA (Alsachim, #6595) alone or in combination with 2.5 μg/mL BDQ, 5 mg/L RIF, 5 mg/L INH, 100 nM BafA1 or 100 nM ConA for an additional 24 h. Samples were fixed, embedded in resin and trimmed as described above before performing acquisition. The sections were imaged by SEM and NanoSIMS[38]. Briefly, 500 nm sections were cut using ultramicrotome EM UC7 (Leica Microsystems, Germany) and mounted on 7 mm × 7 mm silicon wafers. Sections on silicon wafers were imaged using an FEI Verios SEM (Thermo Fisher Scientific, USA) with a 1 kV beam with the current at 200 pA. The same sections were then coated with 5 nm gold and transferred to a NanoSIMS 50 or 50 L instrument (CAMECA, France). The regions that were imaged by SEM were identified using the optical microscope in the NanoSIMS. A focused $^{133}Cs+$ beam was used as the primary ion beam to bombard the sample; secondary ions ($^{12}C^-$, $^{12}C^{14}N^-$, $^{12}C^{15}N^-$, $^{31}P^-$ and $^{79}Br^-$) and secondary electrons were collected. A high primary beam current of ~1.2 nA was used to scan the sections to remove the gold coating and implant $^{133}Cs^+$ to reach a dose of $1 \times 10^{17}$ ions/cm² at the steady state of secondary ions collected. Identified regions of interest were imaged with a ~3.5 pA beam current and a total dwell time of 10 ms/pixel. Scans of 512 × 512 pixels were obtained. Quantification of secondary ion signal intensities was performed using the open source software ImageJ/Fiji v1.53a and the OpenMIMS v3.0.5 plugin. Briefly, bacterial detection and manual segmentation was performed based on the ion signal from $^{31}P$ and corresponding correlative EM images to identify single-bacterial cells. Quantification of intrabacterial PZA/POA levels within region of interests was performed using $^{15}N/^{14}N$ ratio calculated from $^{12}C^{15}N/^{12}C^{14}N$ signals, where the signal from ratiometric images (512 × 512 pixels, 32-bit) defined the ion level of individual bacteria (expressed as parts per $10^4$). The $^{15}N/^{14}N$ ratio value presented is multiplied by 10,000. Background determination of natural $^{15}N$ levels was performed by quantifying $^{15}N/^{14}N$ ratio within untreated or DMSO-treated samples. The obtained control values were tested for normal distribution by performing a quantile–quantile plot followed by a Shapiro–Wilk test and the mean background value ± standard deviation was determined. The statistical empirical rule was used and the mean background value ± 3*standard deviation ($\mu \pm 3*\sigma$) was applied to define a background level with a 99.7%

confidence. All mean $^{15}N/^{14}N$ ratiometric values obtained above this threshold were subsequently considered as PZA/POA positive bacteria. A similar strategy was used to quantify intrabacterial BDQ levels except that detection and quantification was performed using normalised signal from $^{79}Br/^{12}C^{14}N$. Ion values from individual bacteria in each experimental condition were exported as CSV files, imported in the R studio software (The R Project for Statistical Computing, version 1.3.1073) and graphs were plotted with the ggplot2 package (version 3.3.2)

**Correlative light electron ion microscopy (CLEIM)**. Mtb-infected macrophages were treated with 30 µg/mL $[^{15}N_2, ^{13}C_2]$-PZA for 24 h, then washed once with media and stained with 200 nM LysoTracker™ Red DND-99 (Invitrogen, L7528) for 30 min in a humidified 37 °C incubator with 5% $CO_2$. Subsequently, cells were washed twice with 0.2 M HEPES buffer (pH 7.4) and fixed in 0.1% glutaraldehyde (Sigma, G5882), 4% methanol-free paraformaldehyde (Electron Microscopy Sciences, 15710) in 0.2 M HEPES (pH 7.4) for 16–20 h at 4 °C. Fixative was removed and cells were washed three times in 0.2 M HEPES (pH 7.4) before performing DAPI staining for nuclear visualisation. Image acquisition was performed using Leica SP8 confocal microscope equipped with a HC PL APO CS2 63×/1.40 oil objective. Images of 1024 × 1024 pixels were acquired with Diode 405 nm, DPSS 561 nm and HeNe 633 nm lasers where intensities were set up as 2.5%, 5% and 6% respectively. Emitted signal from each channel was collected at 450 ± 30, 585 ± 15 and 710 ± 15 nm for DAPI, LysoTracker and E2-Crimson respectively. Determination of the mean LTR intensity values associated to Mtb in this experiment were performed by manual quantification[42]. Briefly, the two bacterial area of interests were duplicated, and a mask of the bacteria surrounded by a ring of pixel was generated. The bacteria-containing channel was manually thresholded and a single 'Dilate' command was applied to the binary mask in Fiji. This mask was then used to measure the mean fluorescence intensity of pixels in the red fluorescence channel (LTR) in the area using the command 'Measure'. The cells of interest were then resin embedded, the block was trimmed, SEM/NanoSIMS acquisition and analysis was performed as described above. For correlation, fluorescent images, EM and nanoSIMS micrographs were converted to tiff files and linear adjustments made to brightness and contrast using ImageJ/Fiji v1.53a. Images were further aligned to EM micrographs with Icy 2.0.3.0 software (Institut Pasteur, France), using the ec-CLEM Version 1.0.1.5 plugin. No fewer than 10 independent fiducials were chosen per alignment for 2D image registration. When the fiducial registration error was greater than the predicted registration error, a non-rigid transformation (a nonlinear transformation based on spline interpolation, after an initial rigid transformation) was applied as recommended by the software developers[67]. Images were finally displayed using the open source software ImageJ/Fiji v1.53a.

**Cytotoxicity and endolysosomal proteolytic assays**. Approximately $3.5 \times 10^4$ MDM per well were seeded into an olefin-bottomed 96-well plate (Perkin Elmer, 6055302) in complete RPMI medium. After 16–20 h, the medium was replaced by fresh medium containing 100 nM BafA1 or 100 nM ConA for 24–72 h. To evaluate cytotoxicity, cells were stained for 30 min using the Blue/Green ReadyProbes™ Cell Viability Imaging Kit (Invitrogen, R37609) following the manufacturer recommendations. A positive control was done by adding 50 mM hydrogen peroxide (Sigma, 18304) to the staining solution and live-cell imaging was further performed using the OPERA Phenix microscope with a ×40 water-immersion objective. Segmentation and analysis were performed using the Harmony software (Perkin Elmer, version 4.9) where maximum projection of four individual z-planes with an approximate distance of 2 µm was used to perform single-cell segmentation by combining the 'Find nuclei' and 'Find cells' building blocks. Cell viability was determined by counting the number of nuclei in each experimental condition and the proportion of green positive nuclei, where a threshold of $5 \times 10^3$ fluorescent arbitrary units was applied to define positive cells. For endolysosomal proteolytic activity assays, cells were pulsed with 10 µg/ml of DQ™ Red BSA (Invitrogen, D12051) for 4 h[68]. Stained cells were then washed twice and re-incubated in complete RPMI medium containing 1 µg/ml Hoechst 33342 (Immunochemistry Technologies, 639) and live-cell imaging was further performed using the OPERA Phenix microscope with a ×40 water-immersion objective. In these assays, all lasers were set at 20% with an exposure time of 200 ms. After single-cell segmentation, DQ-BSA positive puncta were segmented by using 'Find spot' and their respective properties were determined using the 'Morphology properties' and 'Intensity properties' building blocks. All the results were exported as CSV files, imported in R (The R Project for Statistical Computing) and graphs were plotted with the ggplot2 package

**Live fluorescence imaging and Mtb-LysoTracker co-localisation analysis**. For live-cell imaging, cells were infected with Mtb WT and Mtb ΔRD1 at a MOI of 1 as described above. After, 24 h the culture media was replaced by fresh media only or fresh medium containing 100 nM BafA1 or 100 nM ConA. Approximately 24 h after treatment, infected cells were washed once with PBS buffer (pH 7.4) and stained for 30 min with complete RPMI medium containing 200 nM LysoTracker™ Blue DND-22 (Invitrogen, L7525) in a humidified 37 °C incubator with 5% $CO_2$. Infected cells were then washed with PBS buffer (pH 7.4) and re-suspended in

complete RPMI containing NucSpot® Live 488 (Biotium, #40081) following the manufacturer recommendations. Live-cell imaging was further performed using the OPERA Phenix microscope with a ×63 water-immersion objective and acquisition settings for blue, green and far-red channels were conserved as described above. Segmentation and analysis were performed using the Harmony software (Perkin Elmer, version 4.9) where the E2-Crimson signal from single z-planes was detected using the 'Find Image Region' building block and where a manual threshold was applied to accurately perform bacterial segmentation. Each individual region of interest was transformed into a mask and extended by using the 'Find Surrounding Regions' building block with an individual threshold of 0.8 and including the input region. This mask was used to determine the LysoTracker mean intensity associated to the Mtb region. To determine the percentage of LysoTracker positive bacteria, 200 bacterial regions considered as LysoTracker negative were analysed. The control mean intensity values were tested for normal distribution by performing a quantile–quantile plot followed by a Shapiro–Wilk test and the LysoTracker mean intensity value ± standard deviation was determined. The statistical empirical rule was used and the mean background value ± 3*standard deviation ($\mu \pm 3*\sigma$) was applied to define a LysoTracker negative background level with a 99.7% confidence. All mean fluorescence intensity values obtained above this threshold were subsequently considered as LysoTracker positive bacteria.

**Statistical analysis**. For NanoSIMS experiments statistical analysis was performed using the linear model[38]. Briefly, normalised ion count values were log2 transformed to achieve approximate normality and a (multi-)linear model was applied to predict these log2counts. The following formula was used '(log2(counts) ~ modulator + donor)' allowing the consideration of donor-to-donor variability as parameter. Models were fit using the 'lm()' function in R studio. All p values correspond to the t-statistic assessing significant difference from zero of the model coefficients associated with the reported condition, relative to the baseline control condition. For replication experiments and intracellular antibiotic activity assays, the means between the group of interest were tested for significant differences using one-way ANOVA followed by Tukey's post-test with the 'aov()' and 'TukeyHSD()' functions in R. All the p values contained in the text or the figures are relative to the control condition (unless otherwise stated).

**Reporting summary**. Further information on research design is available in the Nature Research Reporting Summary linked to this article.

## Data availability

All other data supporting the findings of this study are available from the corresponding authors upon reasonable request. Source data are provided with this paper.

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

## Acknowledgements

We would like to acknowledge all members of the Host-Pathogen Interactions in Tuberculosis laboratory for insightful discussions. We thank Luiz Pedro S. de Carvalho for reading the paper and helpful discussions. We are also grateful to the Advanced Light Microscopy and Electron Microscopy facilities from the Francis Crick Institute for their support in various aspects of this study. This work was supported by the Francis Crick Institute (to M.G.G.), which receives its core funding from Cancer Research UK (FC001092), the UK Medical Research Council (FC001092), and the Wellcome Trust (FC001092). For the purpose of Open Access, the author has applied a CC BY public copyright licence to any Author Accepted Manuscript version arising from this submission. H.J. is supported by an Australian Research Council Discovery Early Career Research Award and a Rebecca L Cooper Medical Research Foundation Project Grants. P.S. is supported with a non-stipendiary FEBS long-term fellowship and has received funding from the European Union's H2020 research and innovation programme under the Marie Sklodowska-Curie grant agreement *SpaTime_AnTB* n°892859. The authors are also grateful to the Centre for Microscopy, Characterisation and Analysis at the University of Western Australia, which is funded by the University and both the State and Commonwealth Governments.

## Author contributions

M.G.G. conceived and supervised the project. P.S., D.J.G., H.J. and M.G.G. designed the experiments. P.S., D.J.G., A.F., K.C. and H.J. performed experiments. All authors analysed data and provided intellectual input. P.S. and M.G.G. wrote the paper with input from D.J.G., A.F. and H.J. All authors read the paper and provided critical feedback.

## Competing interests

The authors declare no competing interests.
