## [Peer Review File · Nature Communications]

REVIEWER COMMENTS

Reviewer #1 (Remarks to the Author):

Understanding factors that impact antimicrobial drug susceptibility and resistance is a critical, yet, substantially underexplored area of drug discovery and development. The current manuscript by Santucci and colleagues explores intracellular pharmacokinetics of the first line tuberculosis drug pyrazinamide to understand how cellular microenvironments influence the activity of this important drug. While it has been known for decades that *Mycobacterium tuberculosis* (Mtb) co-opts mildly acidic monocytic phagosomes as a growth niche and that pyrazinamide is only active against Mtb under acidic conditions, no study has documented a direct observation of pH-dependent intracellular bacterium-drug co-localization. Based on the recent observation that antitubercular drug bedaquiline, which is synergistic with many TB drug in vivo, can stimulate autophagy enhancing phagosomal restriction of Mtb, the authors investigate whether the basis for synergy with pyrazinamide relates acidification of Mtb microenvironments. The study presents novel and interesting findings that will be important for TB drug discovery, and presents ideas and approaches that will be useful considerations to the broad field of drug discovery.

Points to consider

Major considerations:

1. The authors acknowledge a major limitation in their study of only being able to detect the pro-drug pyrazinamide and not the active form pyrazinoic acid. As the authors note, there is a long-standing speculation that pyrazinoic acid accumulates within bacilli and perhaps within host compartments due to its charged nature. It is understood that the approach used will not currently permit one to distinguish between these molecules. However, one could use a Mtb strain deficient for pyrazinamidase activity to determine whether lack of this activity alters localization/accumulation. It would be quite interesting, and an important finding, if intramonocytic localization is unchanged, but, intrabacterial localization is dramatically altered.

Minor points:

1. The authors need to provide additional information on the identity of their $^{13}\text{C}_2,^{15}\text{N}_2$ pyrazinamide. Presumably, it is the nitrogens at 4 and 7 positions within the pyrazine ring that are isotopically labeled. Which carbons are isotopically labeled? Also, please describe the source of isotopically labeled pyrazinamide and means of analysis for authenticity.
2. Standard abbreviation for ethambutol is EMB, not ETB. However, since abbreviation is only used once, it should just be deleted.
3. It was not possible to find any description of construction/molecular analysis of the Mtb delta RD1 strain that was used. Reference 40 does not describe construction or validation of this strain, nor does the reference within reference 40. Is there any information that confirms authenticity of this strain?
4. In methods section, "ratio calculated from $^{12}\text{C}^{15}\text{N}/^{12}\text{C}^{14}\text{N}$ signals", should this read " $^{13}\text{C}^{15}\text{N}/^{12}\text{C}^{14}\text{N}$ "?

Reviewer #2 (Remarks to the Author):

Establishing where and to what extent TB drugs accumulate within host cells and in tubercle bacilli residing intracellularly is of fundamental importance for understanding the efficacy of individual drugs in the intracellular environment, how the specific intracellular microenvironments might influence drug MOA, and how different drugs interact with one another when used in combination chemotherapy. Recent advances in imaging have enabled some of these questions to be addressed at single-cell resolution. In a pioneering study published last year, the authors combined

correlated light, electron and ion microscopy to visualize the subcellular localization of bedaquiline (BDQ) and show that this drug accumulated mainly in lipid droplets within host cells, and distributed heterogeneously in bacilli located in different subcellular compartments. In the present study, the authors use a similar experimental approach to investigate the intracellular localisation of PZA, a first-line TB prodrug that synergizes with BDQ. The data reported in this manuscript provide evidence to suggest that the specific microenvironments where PZA and/or its active metabolite, POA, localize, affect both drug accumulation and drug efficacy. The authors further show that the enhancement of PZA accumulation by BDQ occurs via a host-mediated mechanism. Unfortunately, the method used to visualise PZA by ion microscopy, which entailed the incorporation of ¹³C and ¹⁵N isotopic labels in the pyrazine ring moiety of PZA did not allow the prodrug, PZA to be distinguished from its active metabolite, POA.

This is a very well written manuscript that reports some important findings of significant interest to specialist and non-specialist readers, alike. The results are based on impressive quantitative imaging data presented in a set of 5 figures in the main text and 11 others in the SI, which together substantiate the main conclusions drawn from the study. However, the authors should address the following points:

1. The limitation of being unable to distinguish between PZA and POA was acknowledged. However, have the authors considered using a PZA-resistant Δ pncA mutant of Mtb, which is incapable of hydrolyzing PZA to POA, as a comparator strain for analyzing drug accumulation and subcellular distribution? Such a comparative analysis might be especially informative for bacilli residing in host microenvironments that differ in terms of acidity.
2. Pg. 9, bottom of the second paragraph, Figs. 3E and F, and Fig. S9. The authors claim that the differences in efficacy of PZA shown in Figs. 3E and F were confirmed by CFU analysis (Fig. S9). However, whereas PZA at 30 mg/L was efficacious in WT Mtb and reduced the bacterial burden by 40% as ascertained by quantitative imaging (Fig. 3E), this treatment appeared to have no impact on CFU (Fig. S9, panel A). Likewise, while the data shown in Fig. 3 substantiate the following claim: "Notably, although BafA1 greatly reduced PZA efficacy against intracellular bacteria, PZA was still effective against these bacteria and reduced Mtb replication", the data shown in Fig. S9 do not support this. In the case of the Δ RD1 strain (panel B), no significant difference in CFUs was observed between the PZA-treated sample vs. untreated control; on the contrary, the PZA-treated sample showed marginally higher CFUs than the control. Furthermore, some evidence of PZA activity is observed in the WT experiment (panel A); however, is the reduction in CFUs observed in the treated sample vs. untreated control statistically significant? How can the differences in bacterial burden, as determined by quantitative imaging vs. CFU assessment be reconciled?
3. A striking finding is the remarkable heterogeneity in PZA/POA accumulation between host cells and between intracellular bacilli. In the Discussion (pg. 12, second paragraph), the authors postulate that "the phenotypic state of individual cells determines drug uptake and enrichment". How could this hypothesis be tested? Likewise, what are the possible reasons for why "PZA does not accumulate in all intracellular Mtb"? The authors should speculate on these questions. A discussion of the implications of the heterogeneity revealed by quantitative imaging at the single-cell level for the development of new TB drugs and drug combinations where decision-making is reliant on data derived from bulk, population-based analyses would also be helpful.

Reviewer #3 (Remarks to the Author):

The manuscript by Santucci et. al. investigates the subcellular distribution of the anti-TB drug pyrazinamide (PZA) using a comprehensive correlative imaging workflow previously established. The main findings are that substantial accumulation of PZA occurs in a relatively small number of intracellular bacteria, and PZA accumulation for the most part appears to be related to lysosomal integrity, function and/or pH. Co-treatment with bedaquiline dramatically increased PZA accumulation in Mtb, an interesting finding that illuminates the synergistic effect of these antibiotics. The manuscript is well written, the figures are clear, and the findings are of general interest to the community due to the importance of understanding the mechanisms of anti-TB

therapies in relevant contexts. However, there are certain claims that are not fully supported by the data shown, or where alternative hypothesis have not been sufficiently explored, which I believe should be addressed before publication is recommended. In particular, differences in PZA/POA accumulation are mainly ascribed to host cell factors, while bacterial factors such as fitness/viability are not sufficiently investigated.

Major concerns:

1. It is clear from Figures S3 and S5 that the treatment with BafA1 only partially inhibits lysosomal acidification and function. This is not unexpected for such long-term treatments, but it makes it challenging to clearly conclude pH-independent PZA accumulation during BafA1 treatment. The second line of evidence for this conclusion is two bacteria from a single macrophage in a CLEIM experiment (Figure 4). Firstly, although these are indicated as being negative for lysotracker, there is a clear remaining lysotracker signal also in this compartment. In general, lysotracker retention after fixation is not necessarily complete, and it is unclear how the compartments are scored positive or negative. Further, although the complexity and low throughput of these experiments is appreciated, it remains unclear if these bacteria represent unusual outliers or not. I believe the evidence behind pH-independent accumulation of PZA needs to be further substantiated.

2. The increase in PZA accumulation during BDQ treatment is striking. However, BDQ both activates host macrophages and inhibits Mtb growth. The in vitro control experiments do not really clarify which pathway is more relevant, as PZA accumulation and efficacy is mainly an intracellular phenomenon. Further, even though the effects of BDQ are diminished by BafA1, the PZA accumulation during BDQ+BafA1 treatment remains drastically higher than untreated cells. This discrepancy is not discussed. It would greatly strengthen the manuscript if pathways inhibiting Mtb function (e.g. other antibiotics) and activating host macrophages (especially TFEB activation) were independently assessed for their effect on PZA accumulation. The main outstanding issue that should be investigated is if bacterial fitness and/or viability in lysosomes plays a role on PZA accumulation independently of lysosomal pH.

Minor comments:

1. It is not described in the text what the 31P signal represents.

2. Figure 1C: Should show distributions as done for most plots later.

3. p9, from the text: "Interestingly, at a higher concentration (100 mg/L), the inhibition was more efficient but the difference between the Mtb WT and Mtb DRD1 was reduced, suggesting that at higher concentrations PZA might distribute more evenly into more intracellular microenvironments.". Why was it not investigated if this is the case?

4. p9, from the text: "BafA1 treatment reduced PZA efficacy when used at 30 mg/L resulting in a significantly higher bacterial burden, with changes in inhibition values from 40% to 20% for Mtb WT and from 70% to 25% for the Mtb DRD1 mutant (Figure 3F). These differences in efficacy were confirmed by colony-forming units (CFU) analysis (Figure S9).".

It is not entirely clear that Figure S9 actually supports these findings. Rather, it seems from the CFU counts that PZA is ineffective on its own against WT Mtb, and that if anything BafA1 increases the potency of PZA in WT Mtb, although this may be within experimental error.

5. p10, from the text: "These results might also explain why PZA is still very effective against Mtb WT despite its localisation in the cytosol." Typically not all WT Mtb are in the cytosol, so this argument should be clarified.

6. p11, from the text/Figure 5B: "A quantitative analysis of both BDQ and the PZA/POA levels per single-bacterium showed no correlation between the amount of BDQ per bacterium and the increase in PZA/POA ($R=0.14$, $p<0.01$)."

What would be the criteria for a positive correlation here? Judging from the p-value a weak positive correlation seems to be found?

7. p11, from the text/Figure 5C: "High-resolution ion micrographs of 31P, 79Br/12C14N and 14N/15N signal analysis and 3D surface plot reconstruction showed that PZA/POA signal was mainly detected at the periphery of the bacterial cell suggesting a strong association with the bacterial cell wall (Figure 5C-D)."

Can it really be distinguished from these images if PZA/POA is accumulating in the bacterial membrane or rather in the lumen of intact phagolysosomes, but outside the bacteria?

8. p12, from the text: "However, we observed that not all macrophages had detectable levels of PZA/POA and some macrophages were completely devoid of the antibiotic."

These statements seem redundant.

9. PZA has in some reports been shown to be synergistic with for example rifampicin, despite rifampicin not activating macrophages. The implications of these studies should be discussed.

10. The dots in the violin plots appear to be mean values, but this is not clearly stated in the figure legends.

11. It is unclear why standard error of mean rather than standard deviations are displayed in most bar plots. Generally standard deviations should be used when true biological replicates are considered to reflect the donor variability.

Nature Communications Manuscript NCOMMS-20-46835

Point-by-point response to the reviewers' comments

We would like to thank the 3 reviewers for their careful evaluation of our work and insightful suggestions regarding the content of our manuscript.

Please find below a point-by-point response to the reviewer's comments where **our answers are displayed in Blue**, **the actions that have been undertaken and the modifications performed in the new version of the manuscript are indicated in Red** and **some additional references to support our claims are highlighted in Orange**. A complete list of the cited references has been included at the end of our response.

Modifications that have been performed are **highlighted in yellow** in the revised manuscript. The figure numbers and their respective legends have been updated accordingly.

Reviewer #1:

Understanding factors that impact antimicrobial drug susceptibility and resistance is a critical, yet, substantially underexplored area of drug discovery and development. The current manuscript by Santucci and colleagues explores intracellular pharmacokinetics of the first line tuberculosis drug pyrazinamide to understand how cellular microenvironments influence the activity of this important drug. While it has been known for decades that Mycobacterium tuberculosis (Mtb) co-opts mildly acidic monocytic phagosomes as a growth niche and that pyrazinamide is only active against Mtb under acidic conditions, no study has documented a direct observation of pH-dependent intracellular bacterium-drug co-localization. Based on the recent observation that antitubercular drug bedaquiline, which is synergistic with many TB drug in vivo, can stimulate autophagy enhancing phagosomal restriction of Mtb, the authors investigate whether the basis for synergy with pyrazinamide relates acidification of Mtb microenvironments. The study presents novel and interesting findings that will be important for TB drug discovery, and presents ideas and approaches that will be useful considerations to the broad field of drug discovery.

Points to consider

Major considerations:

1. The authors acknowledge a major limitation in their study of only being able to detect the pro-drug pyrazinamide and not the active form pyrazinoic acid. As the authors note, there is a long-standing speculation that pyrazinoic acid accumulates within bacilli and perhaps within host compartments due to its charged nature. It is understood that the approach used will not currently permit one to distinguishing between these molecules. However, one could use a Mtb strain deficient for pyrazinamidase activity to determine whether lack of this activity alters localization/accumulation. It would be quite interesting, and an important finding, if intramonocytic localization is unchanged, but, intrabacterial localization is dramatically altered.

We would like to thank the reviewer for his/her insightful suggestion. We acknowledge that being able to distinguish the prodrug from its active form would be important. Both the development of new isotopically-labelled compounds allowing to discriminate PZA from POA by ion microscopy and the potential use of *pncA* mutant strains were obviously considered as follow up studies in the coming months.

Performing gene inactivation in *M. tuberculosis* requires an extensive genotypic and phenotypic characterisation of the recombinant strain. Because of this, generating a

M. tuberculosis $\Delta pncA$ strain in the allocated timeframe would have been challenging and time-consuming. In order to address this specific point raised by both Reviewer #1 and Reviewer #2, we performed experiments with two additional strains. Both *M. bovis* and *M. bovis* BCG strains, already available in our laboratory, carry a point mutation within their *pncA* genes (C169G) which leads to the His57Asp substitution. The His57 residue being localised within the essential iron binding site of the enzyme, such mutation is detrimental for PncA enzymatic activity and prevents both strains to convert PZA into POA (Scorpio *et al*, 1996, PMID: 8640557; Petrella *et al*, 2011, PMID: 21283666). Moreover, subcellular localisation of these two PZA-resistant strains have been previously characterised by different independent research groups including ours. The vaccine strain *M. bovis* BCG has been previously shown to be exclusively restricted into membrane bound compartments (Van de Wel *et al*, 2007, PMID: 17604718; Simeone *et al*, 2012, PMID: 22319448). Regarding *M. bovis*, our group has recently demonstrated that cytosolic access in human macrophages is restricted (Queval *et al*, 2021, PMID: 33720986).

Our data show that no positive $^{12}\text{C}^{15}\text{N}$ signal enrichment coming from the labelled-PZA was detectable with both *pncA* deficient strains, arguing that conversion of PZA into POA by PncA is required for $^{12}\text{C}^{15}\text{N}$ enrichment. These new results highlight that POA is the main form accumulated within intracellular bacteria in our experimental system (Figure S2, Page 6 & Page 14).

Minor points:

1. The authors need to provide additional information on the identity of their $^{13}\text{C}_2,^{15}\text{N}_2$ pyrazinamide. Presumably, it is the nitrogens at 4 and 7 positions within the pyrazine ring that are isotopically labeled. Which carbons are isotopically labeled? Also, please describe the source of isotopically labeled pyrazinamide and means of analysis for authenticity.

We agree that additional information regarding the ^{13}C and ^{15}N labels could be useful for the readers. Chemical structures of PZA and POA have been added into the *Supplementary information* of the manuscript (Figure S1). Moreover, the labelled [$^{13}\text{C}_2,^{15}\text{N}_2$]-PZA was purchased from Alsachim (<https://www.alsachim.com/fr/>; Reference product #C6595). Chemical purity was assessed by HPLC-UV-MS by the manufacturer and was estimated between 98% and 100%. Isotopic enrichments were estimated around 99% and 98% for ^{13}C and ^{15}N labels respectively. This information was provided and guaranteed by Alsachim in a comprehensive certificate of analysis. Reference product was added in the *Material and Methods* section of the revised manuscript (Page 23).

2. Standard abbreviation for ethambutol is EMB, not ETB. However, since abbreviation is only used once, it should just be deleted.

This abbreviation has been removed from the revised version of the manuscript.

3. It was not possible to find any description of construction/molecular analysis of the Mtb delta RD1 strain that was used. Reference 40 does not describe construction or validation of this strain, nor does the reference within reference 40. Is there any information that confirms authenticity of this strain?

Mycobacterium tuberculosis H37Rv WT and ΔRD1 strains were obtained from William R. Jacobs Jr. (Albert Einstein College of Medicine, New-York, USA), Suzie Hingley-Wilson (University of Surrey, Guilford, UK) and Douglas Young (The Francis Crick Institute, London, UK). The full details regarding the ΔRD1 recombinant strain construction and characterisation are available in the original publication (Hsu *et al*, 2003, PMID: 14557547). This information and additional reference have been added to the revised version of the manuscript (Page 19).

4. In methods section, "ratio calculated from $^{12}\text{C}^{15}\text{N}/^{12}\text{C}^{14}\text{N}$ signals", should this read " $^{13}\text{C}^{15}\text{N}/^{12}\text{C}^{14}\text{N}$ "?

Thanks for pointing this out. Unfortunately, upon the initial set up of this experimental system, we were not able to detect any ^{13}C signal coming from the labelled PZA. Indeed, ^{13}C levels obtained by ion microscopy from PZA-treated and control samples were similar. This is likely due to the matrix effect as mentioned in the first paragraph of the *Results* and the *Discussion* sections. In that context, PZA/POA levels were quantified based on the detectable signal of ^{15}N and thus enrichment was calculated from $^{12}\text{C}^{15}\text{N}/^{12}\text{C}^{14}\text{N}$ signals.

Reviewer #2:

Establishing where and to what extent TB drugs accumulate within host cells and in tubercle bacilli residing intracellularly is of fundamental importance for understanding the efficacy of individual drugs in the intracellular environment, how the specific intracellular microenvironments might influence drug MOA, and how different drugs interact with one another when used in combination chemotherapy. Recent advances in imaging have enabled some of these questions to be addressed at single-cell resolution. In a pioneering study published last year, the authors combined correlated light, electron and ion microscopy to visualize the subcellular localization of bedaquiline (BDQ) and show that this drug accumulated mainly in lipid droplets within host cells, and distributed heterogeneously in bacilli located in different subcellular compartments. In the present study, the authors use a similar experimental approach to investigate the intracellular localisation of PZA, a first-line TB prodrug that synergizes with BDQ. The data reported in this manuscript provide evidence to suggest that the specific microenvironments where PZA and/or its active metabolite, POA, localize, affect both drug accumulation and drug efficacy. The authors further show that the enhancement of PZA accumulation by BDQ occurs via a host-mediated mechanism. Unfortunately, the method used to visualise PZA by ion microscopy, which entailed the incorporation of ¹³C and ¹⁵N isotopic labels in the pyrazine ring moiety of PZA did not allow the prodrug, PZA to be distinguished from its active metabolite, POA.

This is a very well written manuscript that reports some important findings of significant interest to specialist and non-specialist readers, alike. The results are based on impressive quantitative imaging data presented in a set of 5 figures in the main text and 11 others in the SI, which together substantiate the main conclusions drawn from the study. However, the authors should address the following points:

1. The limitation of being unable to distinguish between PZA and POA was acknowledged. However, have the authors considered using a PZA-resistant Δ pncA mutant of Mtb, which is incapable of hydrolyzing PZA to POA, as a comparator strain for analyzing drug accumulation and subcellular distribution? Such a comparative analysis might be especially informative for bacilli residing in host microenvironments that differ in terms of acidity.

We would like to thank the reviewer for his/her constructive comment. A similar comment has been suggested by Reviewer #1 and we have tried to address this point in our new version of the manuscript. Please find our answer above (Reviewer #1 Comment's, Major Consideration) and the corresponding modifications in the revised version of the manuscript (Figure S2, Page 6 & Page 14).

2. Pg. 9, bottom of the second paragraph, Figs. 3E and F, and Fig. S9. The authors claim that the differences in efficacy of PZA shown in Figs. 3E and F were confirmed by CFU analysis (Fig. S9). However, whereas PZA at 30 mg/L was efficacious in WT Mtb and reduced the bacterial burden by 40% as ascertained by quantitative imaging (Fig. 3E), this treatment appeared to have no impact on CFU (Fig. S9, panel A). Likewise, while the data shown in Fig. 3 substantiate the following claim: "Notably, although BafA1 greatly reduced PZA efficacy against intracellular bacteria, PZA was still effective against these bacteria and reduced Mtb replication", the data shown in Fig. S9 do not support this. In the case of the Δ RD1 strain (panel B), no significant difference in CFUs was observed between the PZA-treated sample vs. untreated control; on the contrary, the PZA-treated sample showed marginally higher CFUs than the control. Furthermore, some evidence of PZA activity is observed in the WT experiment (panel A); however, is the reduction in CFUs observed in the treated sample vs. untreated control statistically significant? How can the differences in bacterial burden, as determined by quantitative imaging vs. CFU assessment be reconciled?

This is an important point. We agree that more clarity is required on that paragraph and we agree that the term “confirm” might lead to some misunderstanding. This section was initially written in this format because multiple results obtained by quantitative fluorescence analysis were also obtained in our CFU assay. Statistical analysis was not included in this supplementary figure, because several non-significant differences were obtained when assessed with one-way ANOVA followed by a pairwise t-test. For clarity, all the *p-values* have been now added to the *Supplementary Information (Figure S10)* in the revised version of the manuscript.

Among the interesting similarities, we noticed that Mtb Δ RD1 was more susceptible than its WT counterpart upon PZA treatment suggesting that the results obtained by fluorescence microscopy were reproducible when assessed with an alternative methodology.

Indeed, quantitative imaging showed that Mtb WT replication was slightly but not significantly reduced when treated with 30 mg/L of PZA ($p:0.27$) (**Figure 3**). No significant effect was also observed when comparing PZA efficacy against Mtb WT by CFU counting ($p:0.33$) (**Figure S10**). Regarding Mtb Δ RD1, our fluorescence imaging analysis showed that the replication of the mutant was significantly inhibited in the presence of 30 mg/L of PZA ($p<0.01$) (**Figure 3**). Results obtained by CFU counting (4.02×10^5 CFU/mL vs. 1.26×10^5 CFU/mL) showed a similar pattern with an important inhibition. However, the non-significant *p-value* obtained ($p:0.07$) in this assay, didn't allow us to strongly reject the null hypothesis (**Figure S10**).

In addition, upon Mtb Δ RD1 infection treated with 30 mg/L of PZA, we reported that BAF displays an antagonistic effect ($p<0.05$) which was not observed with the WT strain (**Figure S10**) as determined by fluorescence microscopy (**Figure 3**).

We also would like to raise the fact that despite being the most widely used method to assess intracellular bacterial burden (replication/viability), CFU analysis unfortunately possesses several limitations such as underestimation related to cell death, clumps and aggregates formation upon intracellular growth and/or serial dilution, a high variability and its time-consuming nature (Lerner et al, 2017, PMID: 28242744; Rodriguez et al, 2017, PMID: 29484835; Bussi et al, 2019; PMID: 30916769).

Overall, despite some of the limitations mentioned above, we believe that our results in **Figure S10** do not undermine our findings obtained by high-content fluorescence microscopy in **Figure 3**. For more clarity and avoid any confusion, our claim and conclusions contained within this section have been edited in the revised version of the manuscript (Page 10).

3. A striking finding is the remarkable heterogeneity in PZA/POA accumulation between host cells and between intracellular bacilli. In the Discussion (pg. 12, second paragraph), the authors postulate that “the phenotypic state of individual cells determines drug uptake and enrichment”. How could this hypothesis be tested? Likewise, what are the possible reasons for why “PZA does not accumulate in all intracellular Mtb”? The authors should speculate on these questions. A discussion of the implications of the heterogeneity revealed by quantitative imaging at the single-cell level for the development of new TB drugs and drug combinations where decision-making is reliant on data derived from bulk, population-based analyses would also be helpful.

We also strongly believe that this observation will raise new questions regarding intracellular pharmacokinetics and antibiotic efficacy. This will lead to revisit some of the current concepts applied to treatment of intracellular pathogen infections but will require an in-depth investigation in the years to come.

In that context, “phenotypic state” refers to multiple cellular parameters that may influence the activity of antibiotics but are still uncharacterised. This might include basic morphological properties such as cell size and shape, but also metabolic features such as influx/efflux activities, the global energetic profile (e.g., aerobic, glycolytic, energetic or quiescent), or even the number and activity of key organelles such as lipid droplets or endolysosomes. In this study, we have highlighted that acidification is a major factor contributing to PZA accumulation and efficacy, thus we can postulate that the number of endolysosomes and their acidic/proteolytic content might directly contribute to increase enrichment/efficacy. Investigating such process at the single-cell level will require multiple approaches combining quantitative fluorescence microscopy with specific probes that label host-microenvironment, bacterial fluorescent reporters (providing a read out of antibiotic efficacy) and finally correlate this biological information with NanoSIMS analyses to assess accumulation at the single-cell level.

Regarding single-bacteria heterogeneity, we postulate that numerous features might influence accumulation/efficacy. First, the subcellular localisation and the nature of the microenvironments faced by every bacterium upon intracellular survival/replication appears to us to be critical in this process. In addition, as mentioned in the discussion section of the manuscript, we believe that the dynamics and the time of residence in specific microenvironments by individual bacterium might also greatly contribute to antibiotic enrichment. Finally, all these host-dependent factors probably lead to a wide range of bacterial phenotypes that display different fitness/metabolic profiles, influx/efflux activities or cell-wall integrity for example. But all of these speculations have to be thoroughly investigated in the future.

An additional paragraph in the *Discussion* section of the revised manuscript has been added to discuss some of these emerging concepts (Page 15).

Reviewer #3 (Remarks to the Author):

The manuscript by Santucci et. al. investigates the subcellular distribution of the anti-TB drug pyrazinamide (PZA) using a comprehensive correlative imaging workflow previously established. The main findings are that substantial accumulation of PZA occurs in a relatively small number of intracellular bacteria, and PZA accumulation for the most part appears to be related to lysosomal integrity, function and/or pH. Co-treatment with bedaquiline dramatically increased PZA accumulation in Mtb, an interesting finding that illuminates the synergistic effect of these antibiotics. The manuscript is well written, the figures are clear, and the findings are of general interest to the community due to the importance of understanding the mechanisms of anti-TB therapies in relevant contexts. However, there are certain claims that are not fully supported by the data shown, or where alternative hypothesis have not been sufficiently explored, which I believe should be addressed before publication is recommended. In particular, differences in PZA/POA accumulation are mainly ascribed to host cell factors, while bacterial factors such as fitness/viability are not sufficiently investigated.

Major concerns:

1. It is clear from Figures S3 and S5 that the treatment with BafA1 only partially inhibits lysosomal acidification and function. This is not unexpected for such long-term treatments, but it makes it challenging to clearly conclude pH-independent PZA accumulation during BafA1 treatment. The second line of evidence for this conclusion is two bacteria from a single macrophage in a CLEIM experiment (Figure 4). Firstly, although these are indicated as being negative for lysotracker, there is a clear remaining lysotracker signal also in this compartment. In general, lysotracker retention after fixation is not necessarily complete, and it is unclear how the compartments are scored positive or negative. Further, although the complexity and low throughput of these experiments is appreciated, it remains unclear if these bacteria represent unusual outliers or not. I believe the evidence behind pH-independent accumulation of PZA needs to be further substantiated.

We agree with regards to the partial inhibition of lysosomal function/activity and the recovery that occurs upon long-term exposure to BafA1. During the initial submission of this manuscript, we also tested Concanamycin A (ConA) and found this compound showed a more potent inhibitory effect than BafA1 specially when used for more than 24 hours. Based on these observations and the reviewer's comment, we tested the effect of ConA onto PZA/POA accumulation and efficacy towards both WT and RD1 strains. Our experiments performed in two different donors showed that ConA treatment strikingly inhibits intrabacterial PZA/POA accumulation and efficacy in both WT and RD1 strains. These findings show that long-term treatment with minimal recovery of lysosomal function almost fully counteract PZA antibacterial activity. These complementary results provide further evidence that lysosomal function and acidification is important for PZA/POA enrichment and activity, supporting the idea that a pH-dependent mechanism contributes to PZA/POA accumulation *in cellulo* as suggested by the reviewer. These results have now been included in the *Supplementary information* section of the new version of the manuscript (**Figure S11**) and our statements regarding pH-dependent and pH-independent mechanisms of accumulation/action have been carefully rephrased in the *Results* and *Discussion* sections respectively (Page 4, Page 10 and Page 16).

We also agree that analysis of LysoTracker positivity after fixation can be problematic, especially regarding the well-known retention issues as mentioned by the reviewer. We would like to mention that we have tested numerous lysosomal probes for our correlative approaches and LysoTracker Red DND-99 is still the best one. Determination of the mean LTR intensity values associated to Mtb in this experiment were performed as previously described (Bernard

et al, 2020 PMID: 32938685). Briefly, the two bacterial areas of interest were duplicated, and a mask of the bacteria surrounded by a ring of pixel was generated. The bacteria-containing channel was manually thresholded and a single 'Dilate' command was applied to the binary mask in Fiji. This mask was then used to measure the mean fluorescence intensity of pixels in the red fluorescence channel (LTR) in the area using the command 'Measure'. For clarity, these details have been included in the *Material and Methods* section of the new version of the manuscript (Page 25). Mean fluorescence values from the two compartments of interest show that LTR values were 22.6 and 60.4 AU respectively. Diffuse signal from the cytosol was considered as background and was estimated between 10-20 AU in multiple area picked randomly. Based on these quantitative data, LTR retention was only considered positive for the bacterial area with a value of 60.4 A.U. These values have now been included in the legend of the Figure 4 of the revised version of the manuscript for more clarity (Page 39).

Due to the intricacy and the dynamic nature of our biological system combined to our demanding imaging pipeline, obtaining numerous PZA/POA⁺ independent cells, harbouring both bacteria LTR⁺ and LTR⁻ remains a technical challenge. We agree that the quantification of more events is needed to clearly show that a pH-independent mechanism of PZA accumulation occurs intracellularly. To our knowledge, the data in Figure 4 show for the first time the correlation between a cellular marker, a mycobacteria and antibiotic levels at the subcellular resolution. As mentioned in the *Results* and *Discussion* sections, this correlative approach cannot capture the dynamics of Mtb-LTR association. We are currently expanding our analysis for high-throughput analysis in live-cell in order to count more events, but it remains technically challenging (both the imaging and analysis). We now discuss the limitations of this approach (Page 11).

Overall, we believe that altogether our data show that intracellular pH is an important contributor of PZA/POA accumulation and efficacy and that there are very limited evidences showing that accumulation or efficacy occurs independently of the host endolysosomal pH. We now discussed this to further clarify (Page 4, Page 10 and Page 16).

2. The increase in PZA accumulation during BDQ treatment is striking. However, BDQ both activates host macrophages and inhibits Mtb growth. The *in vitro* control experiments do not really clarify which pathway is more relevant, as PZA accumulation and efficacy is mainly an intracellular phenomenon. Further, even though the effects of BDQ are diminished by BafA1, the PZA accumulation during BDQ+BafA1 treatment remains drastically higher than untreated cells. This discrepancy is not discussed. It would greatly strengthen the manuscript if pathways inhibiting Mtb function (e.g. other antibiotics) and activating host macrophages (especially TFEB activation) were independently assessed for their effect on PZA accumulation. The main outstanding issue that should be investigated is if bacterial fitness and/or viability in lysosomes plays a role on PZA accumulation independently of lysosomal pH.

We agree with the reviewer that the ¹⁵N enrichment upon BDQ-treatment is very high. If our *in vitro* experiments did not allow to directly investigate the molecular bases underlying this potentiation, we strongly believe that this control was needed and raised some interesting interrogations. Indeed, this allowed to exclude that the synergy observed *in cellulo* using concentrations as physiological as possible, doesn't occur independently of the host-cell which agrees with previous observations (Giraud-Gatineau et al, 2020, PMID: 32369020). It also highlighted that *in vitro* (or extracellularly) Mtb growth restriction by BDQ is not a factor that impacts PZA/POA enrichment regardless of the pH.

The effect of BDQ on host macrophage activation status and antibacterial efficacy have been recently shown (Giraud-Gatineau et al, 2020, PMID: 32369020). Using a BDQ resistant Mtb strain, the authors showed that PZA-mediated killing is only enhanced by BDQ intracellularly

via lysosomal activities, and independently of its activity onto its bacterial target, the mycobacterial ATP-synthase (Giraud-Gatineau et al, 2020, PMID: 32369020). To rule out a potentiation effect due to bacterial fitness/viability impairment, we performed NanoSIMS quantification of PZA/POA enrichment upon labelled-PZA/RIF or labelled-PZA/INH co-treatment as suggested by the reviewer. We have now reported in the new version of our manuscript that RIF potentiates PZA/POA accumulation whereas INH does not have any effect onto PZA/POA enrichment in our system. These results strongly suggest that synergistic mechanisms can occur via host-dependent (BDQ) and host-independent mechanisms (RIF). These results also support that high PZA/POA enrichment observed in the context of BDQ, are not directly linked to bacterial viability/fitness as this phenomenon is not observed upon INH treatment. These new results have now been included in the revised version of the manuscript (Figure S14, Page 12 and Page 17).

We agree with the reviewer that the results obtained during the BDQ+BafA1 treatment remains higher than the ones obtained with PZA-only treated cells. This suggests that the BafA1 treatment does not prevent the increase of BDQ-mediated PZA/POA enrichment. Nevertheless, we believe that this is in line with the reviewer first comment suggesting that BafA1 treatment for extensive period does not show potent inhibitory capacity. At the basal level, the use of 100 nM BafA1 results in an important but not complete inhibition of lysosomal function. It is possible that upon BDQ treatment, and subsequently an increase of macrophage lysosomal activity, a higher concentration of BafA1 is required to counteract this overstimulation. In these conditions, obtaining a complete lysosomal inhibition - and complete loss of PZA/POA accumulation- would require a lot of time and optimisation without any guarantee of successful outcome. For clarity, we discussed more in details the limitation regarding of the use of pharmacological inhibitors in the *Results* and *Discussion* sections of the revised manuscript (Page 10 and Page 16).

We considered to experimentally investigate how host-cell activation might affect PZA/POA enrichment. If IFN- γ treatment triggers important restrictive effects in mouse macrophages, numerous independent research groups reported that such effect is not observable in human macrophages, regardless of their differentiation profiles (M-CSF vs GM-CSF) (Douvas et al, 1985, PMID: 3930401; Lerner et al, 2017, PMID: 28242744; Nenasheva et al, 2020 PMID: 32582159; Bernard et al, 2020, PMID: 32938685). Alternatively, an increase in lysosomal biogenesis/activity via TFEB activation could be another possibility which can be achieved by using two distinct experimental strategies. The first one would be to starve the cells in order to trigger the translocation of TFEB from the cytosol to the nucleus, whereas the second option would be to inactivate TFEB expression in human macrophages using siRNA-mediated gene silencing. The first option will result in a pleiotropic metabolic reprogramming which can impact multiple cellular processes and thus preclude to reliably conclude that any accumulation pattern observed (positive or negative) could be directly related to TFEB. The second one, in addition to its challenging nature, recently showed no real effect in resting cells. Indeed, in human monocytes-derived macrophages, no major differences were observed in *atp6v0a1* expression levels or LysoTracker intensity between scramble and TFEB-silenced cells (Giraud-Gatineau et al, 2020, PMID: 32369020) highlighting the technical complexity in addressing these questions in our biological system (human macrophages). Altogether, we believe these experiments are interesting in a different biological *in vitro* system (non-human) in which cytokine stimulation display a clear phenotype or within an experimental model that may facilitate genetic manipulations. Some of these aspects are briefly discussed in the *Discussion* section of the revised manuscript (Page 16).

Minor comments:

1. It is not described in the text what the 31P signal represents.

We thank the reviewer for pointing this out. One sentence with a short description of 31P signal signification has now been included in the revised version of the manuscript (Page 6).

2. Figure 1C: Should show distributions as done for most plots later.

Figure 1C represents the comparison of standard deviation (SD) in $^{15}\text{N}^{12}\text{C}/^{14}\text{N}^{12}\text{C}$ ratiometric signals obtained from individual infected-cells or across the entire biological sample. Thus, each individual cell will have a specific distribution profile of individual bacterial value. Such results obtained from each biological replicate cannot be merged and displayed by using conventional density plots as for the other plots in our manuscript. However, to better illustrate the distribution of the SD in each condition from each biological replicate, the results have been displayed under the form of a box-and-whisker plot linking the results from individual experiments. This new graph has now been included in the new version of the manuscript (Figure 1C).

3. p9, from the text: “Interestingly, at a higher concentration (100 mg/L), the inhibition was more efficient but the difference between the Mtb WT and Mtb DRD1 was reduced, suggesting that at higher concentrations PZA might distribute more evenly into more intracellular microenvironments.”. Why was it not investigated if this is the case?

In this study, we aimed at establishing a new experimental model to visualize PZA/POA in Mtb-infected human macrophages and further understand the cellular pharmacokinetics of PZA/POA using a concentration that is as physiological as possible in our biological system. In that context, we opted for the fixed concentration of 30 mg/L which corresponds to the maximal serum concentration (*C_{max}*) reported in individuals following the standard PZA chemotherapy. Now that this robust experimental system has been set up and initially characterise, numerous investigations will be carried out in our laboratory in a near future. It includes monitoring concentration-dependent effects onto intrabacterial PZA/POA enrichment as suggested by the reviewer but also investigate more in details the temporal dynamics of this accumulation process.

4. p9, from the text: “BafA1 treatment reduced PZA efficacy when used at 30 mg/L resulting in a significantly higher bacterial burden, with changes in inhibition values from 40% to 20% for Mtb WT and from 70% to 25% for the Mtb DRD1 mutant (Figure 3F). These differences in efficacy were confirmed by colony-forming units (CFU) analysis (Figure S9)”. It is not entirely clear that Figure S9 actually supports these findings. Rather, it seems from the CFU counts that PZA is ineffective on its own against WT Mtb, and that if anything BafA1 increases the potency of PZA in WT Mtb, although this may be within experimental error.

We would like to thank the reviewer for raising this point. A similar comment has been suggested by Reviewer #2 and we have tried to address this point in our new version of the manuscript. Please find our answer below (Reviewer #2 Comment's, Major Consideration) and the corresponding modifications in the revised version of the manuscript (Figure S10, Page 10).

5. p10, from the text: “These results might also explain why PZA is still very effective against Mtb WT despite its localisation in the cytosol.” Typically, not all WT Mtb are in the cytosol, so this argument should be clarified.

We understand the reviewer's point. The subcellular localisation of Mtb in our biological system has been previously characterized by transmission electron microscopy and

stereology analysis (Lerner et al, 2017, PMID: 28242744). At 48h post infection around 40% of the bacteria were localised within the cytosol whereas 60% were observed with membrane-bound compartment. This sentence has been changed in the *Results* section of the manuscript and is now clarified in the *Discussion* section (Pages 15-16).

6. p11, from the text/Figure 5B: “A quantitative analysis of both BDQ and the PZA/POA levels per single-bacterium showed no correlation between the amount of BDQ per bacterium and the increase in PZA/POA ($R=0.14$, $p<0.01$).”. What would be the criteria for a positive correlation here? Judging from the p-value a weak positive correlation seems to be found?

We based our conclusions by following the recommendations of the reference paper in medical research (Mukaka, 2012; PMID: 23638278) which indicates that a coefficient correlation obtained within 0-0.3 range is considered as negligible. With a R value of 0.14 and a $p<0.01$, our results clearly show that there is no linear association between our two variables of interest. Our sentence has been rephrased and the reference has now been added in the revised version of the manuscript (Page 12).

7. p11, from the text/Figure 5C: “High-resolution ion micrographs of 31P, 79Br/12C14N and 14N/15N signal analysis and 3D surface plot reconstruction showed that PZA/POA signal was mainly detected at the periphery of the bacterial cell suggesting a strong association with the bacterial cell wall (Figure 5C-D).” Can it really be distinguished from these images if PZA/POA is accumulating in the bacterial membrane or rather in the lumen of intact phagolysosomes, but outside the bacteria?

We fully agree with the reviewer regarding this point. It is impossible for us to clearly state that the signal is coming from the bacterial inner- or outer- leaflet of the cytoplasmic membrane, the mycobacterial cell-wall or the surrounding extrabacterial environment such as the phagosomal lumen for example. Our sentence has been rephrased in the revised version of the manuscript (Page 12) to avoid any misunderstanding.

8. p12, from the text: “However, we observed that not all macrophages had detectable levels of PZA/POA and some macrophages were completely devoid of the antibiotic.”. These statements seem redundant.

This sentence has been rephrased in the new version of the manuscript (Page 14).

9. PZA has in some reports been shown to be synergistic with for example rifampicin, despite rifampicin not activating macrophages. The implications of these studies should be discussed.

As suggested by the reviewer, this experiment has been performed and we have now discussed more details the current knowledge regarding synergistic mechanisms between PZA and other anti-TB drugs including rifampicin. This has now been included in the *Results* and *Discussion* section of our revised manuscript (**Figure S14**, Page 12 and Page 17).

10. The dots in the violin plots appear to be mean values, but this is not clearly stated in the figure legends.

We went through the figure legends and the *Supplementary information* and this information has been added to the new version of the manuscript.

11. It is unclear why standard error of mean rather than standard deviations are displayed in most bar plots. Generally standard deviations should be used when true biological replicates are considered to reflect the donor variability.

We agree that both value range and standard deviation (SD) are commonly used in biological science when displaying results as bar plots because they may reflect how the data are

spread. However, inferential errors such as standard error of mean (SEM) or confidence intervals (CI) are also widely used and recommended when comparing samples from two distinct groups in experimental biology (Cumming et al., 2007, PMID: 17420288). Indeed, SEM allows to measure of how variable the mean will be, if the whole study is repeated many times. Since it's considered appropriate to show inferential error bars such as SEM, and in order to keep consistency all across our study, results were essentially displayed as mean +/- SEM throughout the entire manuscript if not stated otherwise.

List of selected references

Scorpio, A., & Zhang, Y. (1996). Mutations in *pncA*, a gene encoding pyrazinamidase/nicotinamidase, cause resistance to the antituberculous drug pyrazinamide in tubercle bacillus. *Nature medicine*, 2(6), 662–667. <https://doi.org/10.1038/nm0696-662>

Petrella, S., Gelus-Ziental, N., Maudry, A., Laurans, C., Boudjelloul, R., & Sougakoff, W. (2011). Crystal structure of the pyrazinamidase of *Mycobacterium tuberculosis*: insights into natural and acquired resistance to pyrazinamide. *PloS one*, 6(1), e15785. <https://doi.org/10.1371/journal.pone.0015785>

van der Wel, N., Hava, D., Houben, D., Fluitsma, D., van Zon, M., Pierson, J., Brenner, M., & Peters, P. J. (2007). *M. tuberculosis* and *M. leprae* translocate from the phagolysosome to the cytosol in myeloid cells. *Cell*, 129(7), 1287–1298. <https://doi.org/10.1016/j.cell.2007.05.059>

Simeone, R., Bobard, A., Lippmann, J., Bitter, W., Majlessi, L., Brosch, R., & Enninga, J. (2012). Phagosomal rupture by *Mycobacterium tuberculosis* results in toxicity and host cell death. *PLoS pathogens*, 8(2), e1002507. <https://doi.org/10.1371/journal.ppat.1002507>

Queval, C. J., Fearn, A., Botella, L., Smyth, A., Schnettger, L., Mitermite, M., Wooff, E., Villarreal-Ramos, B., Garcia-Jimenez, W., Heunis, T., Trost, M., Werling, D., Salguero, F. J., Gordon, S. V., & Gutierrez, M. G. (2021). Macrophage-specific responses to human- and animal-adapted tubercle bacilli reveal pathogen and host factors driving multinucleated cell formation. *PLoS pathogens*, 17(3), e1009410. <https://doi.org/10.1371/journal.ppat.1009410>

Hsu, T., Hingley-Wilson, S. M., Chen, B., Chen, M., Dai, A. Z., Morin, P. M., Marks, C. B., Padiyar, J., Goulding, C., Gingery, M., Eisenberg, D., Russell, R. G., Derrick, S. C., Collins, F. M., Morris, S. L., King, C. H., & Jacobs, W. R., Jr (2003). The primary mechanism of attenuation of bacillus Calmette-Guerin is a loss of secreted lytic function required for invasion of lung interstitial tissue. *Proceedings of the National Academy of Sciences of the United States of America*, 100(21), 12420–12425. <https://doi.org/10.1073/pnas.1635213100>

Lerner, T. R., Borel, S., Greenwood, D. J., Repnik, U., Russell, M. R., Herbst, S., Jones, M. L., Collinson, L. M., Griffiths, G., & Gutierrez, M. G. (2017). *Mycobacterium tuberculosis* replicates within necrotic human macrophages. *The Journal of cell biology*, 216(3), 583–594. <https://doi.org/10.1083/jcb.201603040>

Rodriguez, D. C., Ocampo, M., Salazar, L. M., & Patarroyo, M. A. (2018). Quantifying intracellular *Mycobacterium tuberculosis*: An essential issue for in vitro assays. *MicrobiologyOpen*, 7(2), e00588. <https://doi.org/10.1002/mbo3.588>

Bussi, C., & Gutierrez, M. G. (2019). *Mycobacterium tuberculosis* infection of host cells in space and time. *FEMS microbiology reviews*, 43(4), 341–361. <https://doi.org/10.1093/femsre/fuz006>

- Queval, C. J., Song, O. R., Carralot, J. P., Saliou, J. M., Bongiovanni, A., Deloison, G., Deboosère, N., Jouny, S., Iantomasi, R., Delorme, V., Debrie, A. S., Park, S. J., Gouveia, J. C., Tomavo, S., Brosch, R., Yoshimura, A., Yeramian, E., & Brodin, P. (2017). Mycobacterium tuberculosis Controls Phagosomal Acidification by Targeting CISH-Mediated Signaling. *Cell reports*, 20(13), 3188–3198. <https://doi.org/10.1016/j.celrep.2017.08.101>
- Bernard, E. M., Fearn, A., Bussi, C., Santucci, P., Peddie, C. J., Lai, R. J., Collinson, L. M., & Gutierrez, M. G. (2020). M. tuberculosis infection of human iPSC-derived macrophages reveals complex membrane dynamics during xenophagy evasion. *Journal of cell science*, 134(5), jcs252973. <https://doi.org/10.1242/jcs.252973>
- Giraud-Gatineau, A., Coya, J. M., Maure, A., Biton, A., Thomson, M., Bernard, E. M., Marrec, J., Gutierrez, M. G., Larrouy-Maumus, G., Brosch, R., Gicquel, B., & Tailleux, L. (2020). The antibiotic bedaquiline activates host macrophage innate immune resistance to bacterial infection. *eLife*, 9, e55692. <https://doi.org/10.7554/eLife.55692>
- Douvas, G. S., Looker, D. L., Vatter, A. E., & Crowle, A. J. (1985). Gamma interferon activates human macrophages to become tumoricidal and leishmanicidal but enhances replication of macrophage-associated mycobacteria. *Infection and immunity*, 50(1), 1–8. <https://doi.org/10.1128/IAI.50.1.1-8.1985>
- Nenasheva, T., Gerasimova, T., Serdyuk, Y., Grigor'eva, E., Kosmiadi, G., Nikolaev, A., Dashinimaev, E., & Lyadova, I. (2020). Macrophages Derived From Human Induced Pluripotent Stem Cells Are Low-Activated "Naïve-Like" Cells Capable of Restricting Mycobacteria Growth. *Frontiers in immunology*, 11, 1016. <https://doi.org/10.3389/fimmu.2020.01016>
- Mukaka M. M. (2012). Statistics corner: A guide to appropriate use of correlation coefficient in medical research. *Malawi medical journal : the journal of Medical Association of Malawi*, 24(3), 69–71.
- Cumming, G., Fidler, F., & Vaux, D. L. (2007). Error bars in experimental biology. *The Journal of cell biology*, 177(1), 7–11. <https://doi.org/10.1083/jcb.200611141>

REVIEWERS' COMMENTS

Reviewer #1 (Remarks to the Author):

I feel the authors have done a nice job addressing my comments. I have no further concerns.

Reviewer #2 (Remarks to the Author):

The reviewers' comments have been satisfactorily addressed. The revisions have enhanced the manuscript which reports findings of particular importance in TB drug discovery and also of general interest to the readership of this journal.

Reviewer #3 (Remarks to the Author):

In response to reviewer comments, the authors have performed additional experiments and clarified several issues in the text. In particular, the additional experiment using alternative phagolysosome-restricted Mycobacteria strains deficient in PncA activity strongly implicate that it is the active POA form of PZA that is visualized in the manuscript. Additional experiments with an alternative lysosomal acidification inhibitor as well as PZA combined with RIF and INH, strengthen the main conclusion that a pH-dependent accumulation of POA to Mtb-containing phagolysosomes is the main mode of PZA action at the single cell level. Therefore, as the authors point out, both host-dependent and Mtb-dependent factors that contribute to altered lysosome environment and altered lysosomal accumulation of Mtb affect PZA efficacy. These findings, together with the establishment of methods to investigate these relationships in a mechanistic manner at the level of single bacteria residing in primary human macrophages, make this work a valuable contribution to understanding context-dependent antibiotic activity. This in turn is crucial to further rational development of novel TB treatment regimens. Other issues and comments raised by the reviewers have also been thoroughly addressed through additional references and clarifications in the text.

I therefore recommend publication of the manuscript by Santucci et.al. in its current form.

A minor note is that the manuscript is lacking a data availability statement (this is only included in the reporting summary). Regarding data availability, I would urge the authors to consider depositing their imaging data in a suitable repository such as the BioImage Archive and EMPIAR for EM data. In this way the sharing of their valuable and unique imaging data with the community would be greatly facilitated.

Sincerely,
Kai S. Beckwith

Nature Communications Manuscript NCOMMS-20-46835A

Point-by-point response to the reviewers' comments

We would like to thank the 3 reviewers for their careful re-evaluation of our work and kind words regarding the content of our study. We are pleased to see that our point-by-point response and revised version of the manuscript have now satisfied all the reviewers' comments.

Reviewer #1 (Remarks to the Author):

I feel the authors have done a nice job addressing my comments. I have no further concerns.

We would like to thank Reviewer #1 for his/her careful evaluation of our manuscript.

Reviewer #2 (Remarks to the Author):

The reviewers' comments have been satisfactorily addressed. The revisions have enhanced the manuscript which reports findings of particular importance in TB drug discovery and also of general interest to the readership of this journal.

We are grateful to Reviewer #2 for his/her insightful inspection of our manuscript.

Reviewer #3 (Remarks to the Author):

In response to reviewer comments, the authors have performed additional experiments and clarified several issues in the text. In particular, the additional experiment using alternative phagolysosome-restricted Mycobacteria strains deficient in PncA activity strongly implicate that it is the active POA form of PZA that is visualized in the manuscript. Additional experiments with an alternative lysosomal acidification inhibitor as well as PZA combined with RIF and INH, strengthen the main conclusion that a pH-dependent accumulation of POA to Mtb-containing phagolysosomes is the main mode of PZA action at the single cell level. Therefore, as the authors point out, both host-dependent and Mtb-dependent factors that contribute to altered lysosome environment and altered lysosomal accumulation of Mtb affect PZA efficacy. These findings, together with the establishment of methods to investigate these relationships in a mechanistic manner at the level of single bacteria residing in primary human macrophages, make this work a valuable contribution to understanding context-dependent antibiotic activity. This in turn is crucial to further rational development of novel TB treatment regimens. Other issues and comments raised by the reviewers have also been thoroughly addressed through additional references and clarifications in the text.

I therefore recommend publication of the manuscript by Santucci et.al. in its current form.

A minor note is that the manuscript is lacking a data availability statement (this is only included in the reporting summary). Regarding data availability, I would urge the authors to consider depositing their imaging data in a suitable repository such as the BioImage Archive and EMPIAR for EM data. In this way the sharing of their valuable and unique imaging data with the community would be greatly facilitated.

Sincerely,

Kai S. Beckwith

We would like to thank Dr. Kai S. Beckwith for his thorough evaluation of our manuscript. The data availability statement has now been included in the revised version of the main manuscript in addition to the reporting summary.

We have now added the Source Data which are provided as a Source Data file and mentioned that all other data supporting the findings of this study are available from the corresponding authors upon reasonable request.